# De-Linearizing Agent Traces: Bayesian Inference of Latent Partial Orders for Efficient Execution

Dongqing Li [* 1]   Zheqiao Cheng [* 2]   Geoff K. Nicholls [1]   Quyu Kong [3]

## Abstract

AI agents increasingly execute procedural workflows as sequential action traces, which obscures latent concurrency and induces repeated step-by-step reasoning. We introduce BPOP, a Bayesian framework that infers a latent dependency partial order from noisy linearized traces. BPOP models traces as stochastic linear extensions of an underlying graph and performs efficient MCMC inference via a tractable frontier-softmax likelihood that avoids #P-hard marginalization over linear extensions. We evaluate on our open-sourced Cloud-IaC-6, a suite of cloud provisioning tasks with heterogeneous LLM-generated traces, and WFCommons scientific workflows. BPOP recovers dependency structure more accurately than trace-only and process-mining baselines, and the inferred graphs support a compiled executor that prunes irrelevant context, yielding substantial reductions in token usage and execution time.

## 1. Introduction

Large language model (LLM) agents are increasingly deployed for multi-step procedural tasks, yet their execution remains inefficient and unreliable. A common design pattern treats each decision step as a fresh planning problem, repeatedly invoking expensive reasoning even for tasks that have been successfully executed many times before. This repeated re-planning not only incurs substantial computational cost (Zhang et al., 2025; Gao et al., 2025), but also increases exposure to stochastic execution errors such as hallucinated actions or invalid plans (Valmeekam et al., 2023). These issues suggest that reliable autonomy requires mechanisms for *reusing* previously successful procedural structure, rather than re-deriving it from scratch at every execution.

In this work, we propose to recover and reuse such structure by explicitly modeling the latent dependencies underlying agent executions. We introduce Bayesian Partial Order Planning (BPOP), a probabilistic framework that infers an explicit procedural graph from historical agent traces. BPOP treats execution logs as noisy observations of an underlying partial order over actions, so that necessary precedence constraints can be separated from incidental execution order. Figure 1 gives a running Cloud-IaC example. It shows how differently ordered successful traces arise from the same latent partial order, and how this recovered structure is compiled into a reusable graph executor.

The core algorithmic challenge is that probabilistic inference over partial orders is notoriously difficult; marginalizing over all valid linear extensions is #P-complete (Brightwell & Winkler, 1991). We avoid this bottleneck with a tractable frontier-softmax likelihood, which scores local choices among currently feasible actions. The inferred posterior structure can then be compiled into a lightweight executor that restricts action selection to feasible frontiers, reducing unnecessary reasoning and execution variance.

**Contributions.** We build on prior Bayesian partial-order inference, including latent representations that induce priors over posets. Our novelty lies in extending this perspective to agent execution traces and executable workflow compilation. **(1) Agent traces as latent workflow linearizations.** We formulate repeated agent executions as noisy linearizations of an underlying partial-order workflow, separating incidental execution order from necessary precedence constraints. **(2) Frontier-softmax inference.** We introduce a tractable frontier-softmax likelihood that scores choices among currently feasible actions, avoiding explicit marginalization over all linear extensions while supporting efficient posterior inference. **(3) Recoverability, feasibility, and execution.** We analyze both structural recovery and execution feasibility, showing when trace diversity is sufficient to recover the latent graph and when the inferred graph still preserves the dependencies needed for valid execution. **(4) Graph compilation.** We compile the inferred partial order into a frontier-based executor that reduces repeated LLM reasoning and improves execution efficiency on scientific-

---
[*]Equal contribution  [1]Department of Statistics, University of Oxford, Oxford, United Kingdom [2]Zhejiang University, Hangzhou, China [3]Independent Researcher, Hangzhou, China. Correspondence to: Dongqing Li <dongqing.li@kellogg.ox.ac.uk>.

*Proceedings of the $43^{rd}$ International Conference on Machine Learning*, Seoul, South Korea. PMLR 306, 2026. Copyright 2026

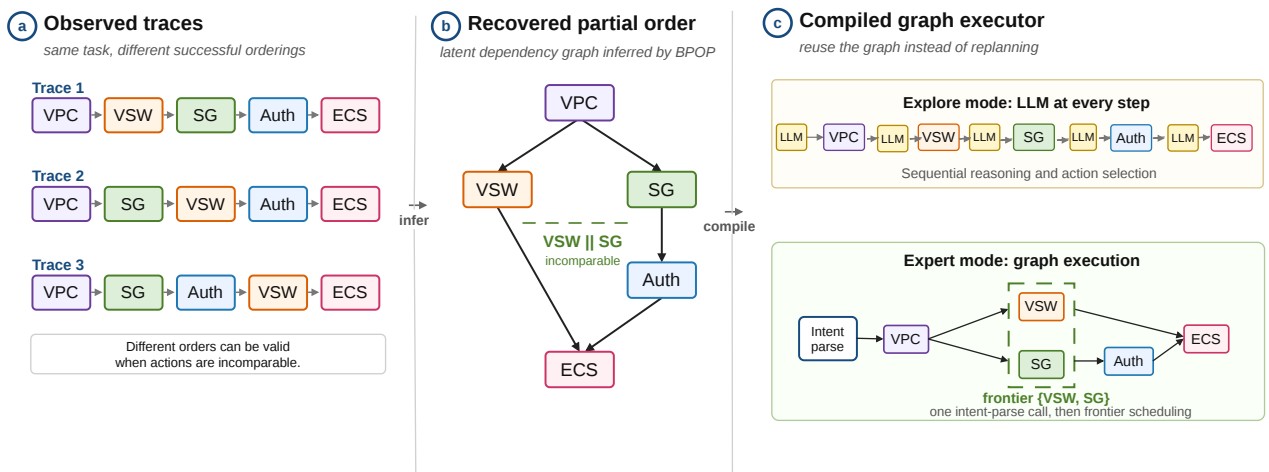

*Figure 1.* **Running Cloud-IaC example.** Successful agent executions are observed as linear traces, but different traces may order independent actions differently. BPOP treats these traces as noisy linearizations of a latent partial-order workflow. Here, after `CreateVPC`, `CreateVSwitch` and `CreateSG` are incomparable; `AuthorizeSG` depends on `CreateSG`; and `RunInstances` depends on both `CreateVSwitch` and `AuthorizeSG`. The inferred graph can then be compiled into a frontier-based executor that replaces repeated step-by-step LLM planning with graph execution.

workflow and Cloud-IaC tasks. Code is available online.[1]

## 2. Background and Problem Formulation

Our objective is to infer an *interpretable, executable action program* in the form of a partial order, which is a DAG.

**Preliminaries (partial orders).** Let $\mathcal{A} = \{1, \ldots, m\}$ denote the set of actions with size $m$. A (strict) partial order is a binary relation $\succ$ on $A$ that is (i) *irreflexive* ($i \nsucc i$) and (ii) *transitive* ($i \succ j \land j \succ k \Rightarrow i \succ k$). We represent $\succ$ by an adjacency matrix $h \in \{0, 1\}^{m \times m}$ where $h_{ij} = 1 \iff i \succ j$. The matrix $h$ encodes all implied dependencies, and the *transitive reduction* of $h$ yields the DAG cover (Hasse diagram Figure 2 left) with no redundant edges (See the full preliminary in Appendix A).

**Action precedence model.** We consider tasks defined over a finite set of atomic actions $\mathcal{A}$ (Appendix D.1). Rather than modeling a state-dependent policy $\pi(a_t \mid s_t)$, we operate in a *state-free* regime and assume access only to execution traces. We posit a latent strict partial order $h = (\mathcal{A}, \succ_h)$, where $a_i \succ_h a_j$ denotes a necessary precedence constraint and $a_i \parallel_h a_j$ denotes potential concurrency.

We do not observe $h$ directly. Instead, we observe a dataset $\mathcal{D} = \{y^{(1)}, \ldots, y^{(n)}\}$ of $n$ successful execution logs, where each trace $y^{(i)} \in \mathcal{D}$ is a total order over a subset of $\mathcal{A}$. Those heterogeneous traces are from different LLMs completing the same task. We treat each trace as a (possibly noisy) *linear extension* of the latent partial order (Figure 2 right;

[1] https://github.com/hollyli-dq/po_inference_agent

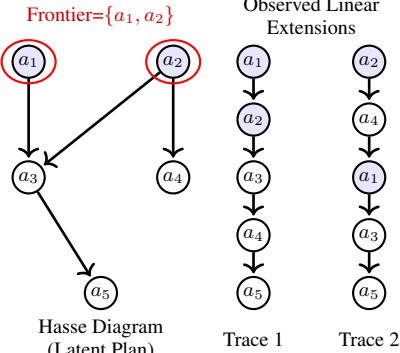

*Figure 2.* **From Chaos to Structure.** BPOP infers the latent Hasse diagram (Left) from diverse linear traces (Right), identifying that $a_1$ and $a_2$ are concurrent (Frontier).

definitions are in Appendix A).

**Structural Coverage (IP-Cov).** Concurrency is identifiable only when traces exhibit sufficient variation to rule out fixed dependencies. We define IP-Cov as the fraction of ground-truth incomparable pairs $\mathcal{P} = \{(i, j) : h_{ij} = h_{ji} = 0\}$ that are witnessed in *both* relative directions ($a_i \succ a_j$ and $a_j \succ a_i$). This metric acts as a quantifiable proxy for trace diversity (formal definition in Appendix E.1). For the practical data acquisition strategy used to maximize this coverage in the absence of ground truth, see Appendix E.2.

**Inference Problem.** We formulate the recovery of agent control structure as Bayesian rank aggregation (Nicholls et al., 2025). The environment arbitrarily serializes concurrent actions, conflating essential precedence constraints with incidental ordering. Successful execution traces $\mathcal{D} =$

$\{y^{(1)}, \ldots, y^{(n)}\}$ are *linear extensions* of an unknown true partial order $h^\star$ expressing precedence. The Bayes posterior summarizes these data and allows us to compute estimators $\widehat{h}$ minimizing the Bayes risk for $h^\star$, disentangling true order-dependencies from random serialization effects.

Importantly, $\widehat{h}$ is not only a statistical estimate but an *operational abstraction*. When compiled into a frontier-based execution engine, it defines a deterministic and parallelizable control policy that replaces repeated per-step LLM planning in routine settings. Our experiments therefore evaluate a dual claim: (i) that partial-order structure is statistically identifiable from traces, and (ii) that improvements in structural recovery yield measurable reductions in runtime reasoning cost during execution.

**Problem Scope: Convergent Procedural Tasks.** BPOP targets convergent domains governed by stable dependencies (e.g., cloud provisioning). We treat execution traces as *unrolled* acyclic graphs, where repeated actions map to distinct occurrences ($A_1 \rightarrow B \rightarrow A_2$). This formulation allows us to distill strict partial orders from cyclic agent policies. It is particularly valuable for enterprise SOP automation—such as Customer Relationship Management (CRM) lead qualification, Enterprise Resource Planning (ERP) ticket resolution, or user onboarding. While Hong et al. (2024) demonstrated that SOPs significantly improve agent reliability, our approach models these SOPs from traces, essential for amortizing the high cost of inference in production.

# 3. Methodology: Bayesian Partial Order Planning Model

We formulate *de-linearizing* agent traces as a Bayesian structure learning task. Rather than imposing rigid graph constraints that limit expressivity, we model the underlying task logic using a continuous latent space representation.

Our prior over partial orders is a variant (Nicholls et al., 2025) of a random order (Winkler, 1985). Each atomic action $a_j \in \mathcal{A}$ is associated with a $K$-dimensional latent vector $U_j \in \mathbb{R}^K$ (See Figure 3). The discrete partial order $h = (\mathcal{A}, \succ_h)$ is induced by component-wise dominance across these $K$ latent dimensions:

$$a_i \succ_h a_j \iff U_{i,k} > U_{j,k} \quad \forall k \in \{1, \ldots, K\}. \quad (1)$$

This formulation interprets a partial order as the intersection of $K$ total orders (realizers). As the dimensionality $K$ increases, the model gains the capacity to represent any finite partial order (Dushnik & Miller, 1941), allowing it to capture complex, overlapping dependencies that simpler tree-based models often miss.

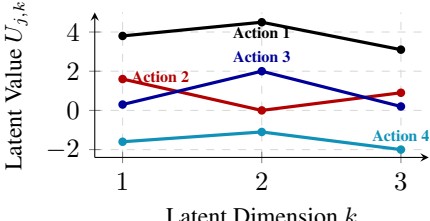

*Figure 3.* **Latent Representation of Partial Order.** Action 1 strictly dominates all others ($1 \succ \{2, 3, 4\}$). Actions 2 and 3 intersect across dimensions, encoding concurrency ($2 \parallel 3$).

## 3.1. The Prior over latent embeddings

For a scenario with $m$ actions and latent dimension $K$, we place a Gaussian prior over action embeddings:

$$U_j \sim \mathcal{N}(\mathbf{0}, \Sigma_\rho) \quad (2)$$
$$h = \text{Dom}(U) \quad (3)$$

**Gaussian prior (Eq. 2)** The matrix $U \in \mathbb{R}^{N \times K}$ parameterizes preference relations over actions. Each action vector $U_j$ is drawn from a zero-mean multivariate normal distribution, independently for each $j$. The covariance matrix $\Sigma_\rho \in \mathbb{R}^{K \times K}$ has a simple exchangeable form (e.g., $\Sigma_\rho = (1 - \rho)I + \rho \mathbf{1}\mathbf{1}^\top$ with $\rho \in [0, 1]$). We optionally infer $\rho$ and $K$ (via a truncated Poisson prior on $K$) to control the depth and complexity of the poset.

**Latent dependency structure (Eq. 3):** The dominance operator $\text{Dom}(\cdot)$ serves as the bridge between the continuous latent space and the discrete graph topology. It deterministically maps the agent's vector matrix $U$ to a partial order $h$ via the intersection rule defined in Eq. (1): a directed edge $a_i \rightarrow a_j$ exists if and only if action $i$ dominates action $j$ across all $K$ dimensions(See Figure 3).

## 3.2. Generative Process: Frontier and Noisy Execution

Given a latent dependency structure $h$ (a strict partial order over action instances), each observed trace $y^{(i)} = (y_1, \ldots, y_T)$ is modeled as the outcome of a sequential execution process (linear extension) constrained by $h$: hard precedence constraints are respected, while the ordering of concurrent actions is resolved through agent decision-making rather than arbitrary serialization. This formulation separates true dependency relations from incidental serializations due to single-threaded execution or logging artifacts. We propose the likelihood as a Plackett–Luce (stagewise MNL, (Luce et al., 1959) model restricted to the poset frontier. This is a Plackett–Luce model with a *state-dependent* choice set or Boltzmann distribution (Ziebart et al., 2008) over the set of topologically feasible actions:

**Feasibility via the frontier.** Given an observed trace $y$, let $y_{<t} = (y_1, \ldots, y_{t-1})$ denote its prefix. Given a partial order

$h = (\mathcal{A}, \succ_h)$, define the set of remaining (not-yet-executed) actions at time $t$ as $R_t \triangleq \mathcal{A} \setminus \{y_1, \ldots, y_{t-1}\}$. The feasible set at time $t$ is the *frontier*, i.e., the set of minimal elements of $R_t$ under $\succ_h$:

$$\mathcal{F}_t(h; y_{<t}) \triangleq \{a \in R_t \ : \ \nexists b \in R_t \text{ such that } b \succ_h a\}. \tag{4}$$

Equivalently, $a \in \mathcal{F}_t$ iff all of its prerequisites under $h$ have been completed. Any linear extension consistent with $h$ must satisfy $y_t \in \mathcal{F}_t(h; y_{<t})$ for every step $t$.

**Frontier-softmax likelihood.** An agent working from partial order $h$ selects action $y_t$ from the current frontier $\mathcal{F}_t(h)$ (Eq. 4) with probability weighted by *successor utility* $Q(y_t; h, t)$. The conditional probability it selects $y_t$ next is:

$$p(y_t \mid y_{<t}, h) \ = \ (1 - \epsilon) \, \frac{\exp(\beta \, Q(y_t; h, t))}{\sum_{a \in \mathcal{F}_t(h)} \exp(\beta \, Q(a; h, t))} + \frac{\epsilon}{|R_t|}, \tag{5}$$

where the first term is set equal 0 when $y_t \notin \mathcal{F}_t(h)$.

Here, $\beta > 0$ is the *inverse temperature*, controlling how sharply the policy concentrates on high-utility actions (Sutton & Barto, 2018). This parameter allows the model to adapt to agents of varying rationality. The parameter $\epsilon$ introduces a "trembling-hand" component (Selten, 1975), mixing the rational frontier choice with a uniform distribution over all remaining actions. This regularization ensures the likelihood remains strictly positive when logging latency causes the observed trace to violate the partial order $h$.

The frontier-softmax likelihood is a tractable inductive bias, not a literal model of agent policy. We do not assume that agents explicitly optimize descendant count; the utility simply favors feasible actions that unlock downstream work, which is a reasonable surrogate in structured workflows where early failure of prerequisite or bottleneck actions can make later work unnecessary. This avoids marginalizing over all linear extensions while preserving the execution constraint each next action must lie in the feasible frontier.

**Successor Utility.** The utility $Q$ is a topological heuristic based on *descendant cardinality*. We score each feasible action $a \in \mathcal{F}_t$ by the size of its reachable subgraph in the latent partial order $\widehat{h}$. This policy prioritizes bottleneck actions that are the prerequisites for the largest number of future actions. Let $S_t(a)$ be the descendant-count at step $t$:

$$S_t(a) \triangleq |\{ b \in R_t : a \succ_h b \}|,$$

$$Q_{\text{succ}}(a; h, t) = \begin{cases} \log(1 + S_t(a)), & a \in \mathcal{F}_t(h), \\ -\infty, & \text{otherwise.} \end{cases} \tag{6}$$

The transformation $\log(1 + S_t(a))$ imposes diminishing returns for massive subgraphs to prevent them from dominating the probability distribution, while ensuring a defined score for leaf nodes (where $S_t(a) = 0$). See Appendix C.2 for a visual breakdown of this stepwise likelihood.

A generic trace $y$ is built sequentially with likelihood

$$p(y \mid h, \beta, \epsilon) \ = \ \prod_{t=1}^{T} p(y_t \mid y_{<t}, h, \beta, \epsilon). \tag{7}$$

**Tractability.** Standard likelihoods that marginalize over linear extensions face #P-complete counting complexity. BPOP sidesteps this by decomposing the trace into sequential local choices from the feasible frontier (Eq. 5). As derived in Appendix C.1, this formulation allows likelihood evaluation in polynomial time $\mathcal{O}(|\succ_h| + T|\mathcal{A}|)$ rather than factorial time. This computational efficiency renders full posterior inference practical for long execution logs.

### 3.3. Posterior Inference

We infer the latent SOP structure and hierarchical parameters in a Bayesian framework. Let $\mathcal{D} = \{y^{(i)}\}$ denote the set of successful traces. The unknowns are the embedding $U \in \mathbb{R}^{m \times K}$, $\rho$, the inverse temperature $\beta > 0$ and optionally the latent dimension $K$. The posterior is

$$p(U, \rho, \beta, K \mid \mathcal{D}) \ \propto \ p(U \mid \rho, K) \, p(\rho) \, p(\beta) \, p(K)$$
$$\times \prod_i p\big(y^{(i)} \mid h(U), \beta, \epsilon\big). \tag{8}$$

Here, $p(y^{(i)} \mid h(U), \beta, \epsilon)$ is the frontier-softmax likelihood (Eq. 7); we treat the slip rate $\epsilon$ as a fixed hyperparameter to ensure numerical stability. The prior for $U$ is given in Eqs. 2–3 and for $\rho$, $\beta$ and $K$ in Appendix B. For inference, we employ a Metropolis-within-Gibbs sampler, incorporating reversible-jump moves for $K$ and a dimension-cycling proposal scheme. MCMC details are given in Appendix B.

**Poset Point Estimation** We summarize the posterior distribution over partial orders by computing marginal edge probabilities $\hat{\pi}_{ij} = \frac{1}{T} \sum_{t=1}^{T} \mathbb{I}\big(i \succ_{h^{(t)}} j\big)$. Let $h^\star$ be the unknown true partial order. We compare two strategies for estimating $h^\star$: (A) the **Marginal Threshold Estimator** $(\hat{h}_{\text{thr}}(\alpha))$, where $\alpha \in (0, 1)$ and $i \succ_{\hat{h}_{\text{thr}}} j \ \Leftrightarrow \ \hat{\pi}_{ij} \geq \alpha$ characterizes the trade-off between precision and recall (motivated by asymmetric decision costs), and (B) the **Marginal Mode Estimator** $(\hat{h}_{\text{mode}})$, which selects the relation type $(i \succ j, j \succ i$, or incomparability $i \parallel j)$ with the highest posterior mass (Bayes-optimal under 0–1 Hamming loss).

### 3.4. Evaluation for Recovery

We evaluate recovery of the ground-truth partial order/SOP $h^\star$. Let $\widehat{h}$ be the estimated partial order $(\hat{h}_{\text{thr}}(\alpha)$ or $\hat{h}_{\text{mode}})$ and let $\text{TR}(\widehat{h})$ and $\text{TC}(\widehat{h})$ be the inferred transitive reduction and closure respectively. We report Precision/Recall/F1 for graph edges. Crucially, the cost of structural errors in unsupervised execution is asymmetric. While preference-based tasks might tolerate edge reversals, in our simulation

environments, such violations are catastrophic. A **False Positive** dependency merely reduces parallelism (a minor efficiency penalty), whereas a **False Negative** (missing a constraint) triggers premature execution and runtime crashes.

Additionally, we report two diagnostics that directly impact action ranking at execution time: **Feasibility**, the fraction of observed successful traces that remain linear extensions of $\mathrm{TC}(\widehat{h})$ (detects over-constraint that would incorrectly prune the frontier), and **IP-Cov**, the fraction of ground-truth incomparable pairs witnessed in *both* orientations across traces (See Appendix E.3)

## 4. From Structure to Efficient Execution

MCMC inference is a one-time offline cost; the learned partial order is reused across executions, yielding negligible amortized planning cost and substantial token savings. We translate the inferred posterior into a deterministic *Graph Execution Engine* (GEE, See the detailed design in Appendix D.2), which operates within a **Tri-Modal Framework** (See Figure 11) to balance efficiency and robustness. We further propose metrics to evaluate the efficiency.

### 4.1. Tri-Modal Execution

**The Tri-Modal framework.** To ensure task success under varying conditions, the system dynamically switches execution across three operating modes as detailed in Section D.3.

- **EXPERT (GEE-Only):** Executes the compiled SOP deterministically. The inferred SOP specifies control flow, but execution additionally requires data flow (how parameters propagate across tool calls ).

- **HYBRID (GEE + Fallback):** Prioritizes the GEE but safeguards against compilation errors. If the GEE encounters a fault (e.g., missing blackboard inputs due to a missing edge in $\widehat{h_\tau}$), control reverts to an LLM planner for recovery.

- **EXPLORE (LLM-Only):** Agent framework to generate and collect diverse traces for experiment scenarios.

**The GEE.** The GEE executes using a frontier-based scheduler and shared data IO blackboard (Details in Appendix D.2). To build the GEE, we compute $\widehat{h} = \hat{h}_{\mathrm{thr}}(\alpha)$. The threshold $\alpha$ is a **Risk–Efficiency Knob**: higher $\alpha$ yields sparser graphs with greater concurrency but higher risk of missing dependency bugs; lower $\alpha$ adds more edges, over-specifying for safety at the cost of parallelism. In experiments, we tune $\alpha$ to maximize the structural F1-score, comparing $\hat{h}_{\mathrm{thr}}(\hat{\alpha})$ against the mode-estimator $\hat{h}_{\mathrm{mode}}$.

In applications, with $h^\star$ unknown, we set $\alpha = 1/3$. This prioritizes dependency recall, preferring slight over-

serialization to avoid catastrophic failures from missing critical edges. The posterior probabilities for relation types $i \succ j$, $j \succ i$, $i \parallel j$ sum to one, so $\hat{h}_{\mathrm{thr}}(1/3) \simeq \hat{h}_{\mathrm{mode}}$, hence $\hat{h}_{\mathrm{thr}}$ is close to Bayes optimal for the 0-1 loss for action precedence. In practice it is actually more useful, because it hedges slightly against missing critical edges.

### 4.2. Evaluation for Execution Efficiency

We evaluate *operational impact*, asking whether higher-quality recovered structure leads to more efficient and reliable execution. When executing $\widehat{h}$ with frontier scheduling, we report: (i) *Success rate*, the fraction of tasks completing without API errors; (ii) *Completeness rate*, the fraction executing all expert-required actions; (iii) *Fallback rate*, the fraction of tasks or actions triggering LLM reasoning; (iv) *LLM calls/task*, the average number of reasoning steps; and (v) *Tokens/task*, total LLM token consumption. See Table 7 for formal definitions.

## 5. Experiments

Our experiments evaluate (i) structural recoverability of partial orders (ii) their downstream execution utility. We assess recoverability on two controlled benchmarks, and evaluate execution efficiency on a single realistic agent workflow where the inferred structure is compiled for execution.

### 5.1. Datasets

**WFCommons Workflows.** We validate on open and reproducible scientific workflows from the WFCommons WfInstances corpus (Coleman et al., 2022), which provides real workflow execution instances in WfFormat JSON, including per-task timing and dependency information. We include SRASearch (WfCommons Project, 2021), a 22-task Pegasus fork–join bioinformatics workflow with 5 observed executions, and Epigenomics (Juve et al., 2013), a larger Pegasus workflow for paired-end read alignment and variant calling with a mid-sized DAG of 41 tasks and a richer parallel structure. For each instance, we treat the workflow specification DAG as ground truth and the execution logs as observed linearizations. Appendix F.2.1 gives preprocessing details.

**Cloud Agent Provisioning.** The full Cloud-IaC-6 benchmark has been open-sourced and anonymized for the review process from an internal *agent-based cloud management platform* on the cloud platform, where an autonomous agent interprets a high-level user query and incrementally provisions the required resources (Authors, 2025). The scenarios span simple virtual networking to complex high-availability clusters and range from 5–12 nodes (Appendix F.3.1), covering heterogeneous resource types including networking, compute, storage, and load balancing (see Table 13 for product definitions). Ground-truth dependency graphs were man-

ually specified and validated by cloud architects. We consider two complementary trace sources: (1) *LLM-generated traces* ($n = 54$) from diverse agents (Qwen, DeepSeek), capturing realistic variation in planning and action ordering (Trace Example Figure 15); and (2) *synthetic traces* sampled from the ground-truth graphs with controlled noise to vary trace informativeness (IP-Cov $\in [0.5, 1.0]$).

## 5.2. Baselines.

We compare against four baselines. (i) *Majority*, which infers a precedence constraint $i \succ j$ whenever the *empirical* precedence $\hat{p}(i \succ j) > 0.5$ in the trace data, followed by greedy cycle breaking and projection to a DAG cover. We add two process-mining baselines: (ii) *Inductive Miner* (IMf) (Leemans et al., 2013) and (iii) *Heuristics Miner* (Weijters et al., 2006), which extract precedence constraints from discovered process models and are similarly projected to DAG covers for evaluation. See Appendix F.1 for algorithmic descriptions of these methods. Finally, we compare with a *Bayesian Queue-Jump* (QJ) baseline (Nicholls et al., 2025) (Appendix F.1.5), where trace likelihoods depend on the number of linear extensions (NLE) of the candidate poset, a #P-complete problem (Brightwell & Winkler, 1991). This is a Bayesian baseline for our frontier-softmax likelihood.

## 5.3. Computational Efficiency and Scalability

BPOP is an offline workflow-compilation step, not a per-execution planner. Its inference cost is paid once to recover a reusable dependency graph from historical traces, and is amortized over future executions where dependency recovery can improve reliability, observability, and efficiency. Our implementation stores candidate dependencies as a binary adjacency matrix, requiring $O(|\mathcal{A}|^2)$ worst-case space. This remains quadratic asymptotically, but the matrix can be represented using bit-packing or sparse bitsets, so memory is not the bottleneck at the workflow scales studied.

This paper targets single structured workflows with moderate action spaces. For larger or longer-horizon tasks, natural extensions include hierarchical partial orders that decompose workflows into reusable subtasks, and variational or other approximate posterior methods to reduce inference time while retaining uncertainty estimates.

## 5.4. WFCommons Results

For WFCommons experiments, we run reversible-jump MCMC for 1M iterations per graph, with individual runs taking 2 hours (22 nodes) or 4.5 hours (41 nodes) on a single CPU core and trivially parallelizable across workflows and IP-Cov settings. Threshold selection is discussed in Appendix F.2.5: the simpler SRASEARCH favors a conservative threshold ($\alpha = 0.5$) for precision, whereas the

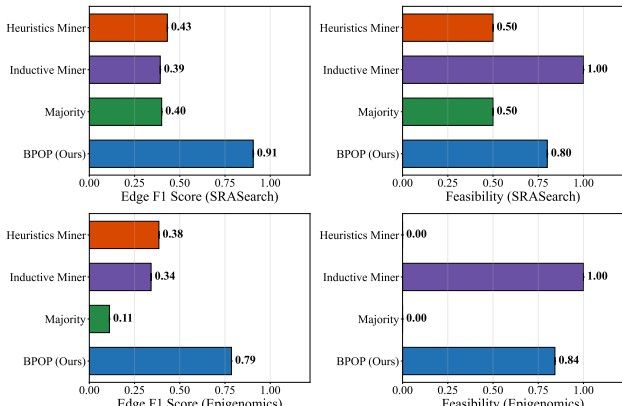

*Figure 4.* **Scientific Workflow Performance (IP-Cov=0.95).** Top: SRASEARCH; Bottom: EPIGENOMICS. BPOP maintains both high Edge-F1 and feasibility, whereas baselines fail to produce executable graphs on the more complex workflow.

highly parallel EPIGENOMICS benefits from the theoretical baseline ($\alpha = 1/3$) to recover concurrent branches. This baseline is uniformly optimal or near-optimal (see Tables 11 and 16) and is recommended for use when ground truth is not available. In contrast, the Bayesian Queue-Jump (QJ) baseline counts NLEs; empirical profiling on SRASEARCH predicts runtimes exceeding **1,000 hours**, so QJ is infeasible at WFCommons scale (See Appendix F.2.3, F.2.4)

**Structural Recovery and Execution Validity.** As shown in Figure 4, BPOP consistently outperforms all baselines in recovering ground-truth structure while maintaining executability. On SRASEARCH, BPOP achieves an Edge-F1 of $0.91$, substantially exceeding the strongest baseline (Heuristics Miner: $0.43$), and on the more complex EPIGENOMICS pipeline it attains an Edge-F1 of $0.79$, compared to $0.38$ for Heuristics Miner and $0.11$ for Majority. BPOP remains robust under data scarcity: even at IP-Cov $\approx 0.6$, it maintains meaningful accuracy, whereas baselines degrade sharply and often require near-complete observation of pairwise orderings. This improved structural stability does not come at the cost of executability. On EPIGENOMICS, Majority and Heuristics Miner collapse to zero feasibility due to over-constraining the graph, while BPOP maintains robust feasibility ($0.84$). Inductive Miner achieves perfect feasibility ($1.00$) but does that by producing overly permissive "flower models" with poor structural fidelity (Edge-F1 $\leq 0.39$).

**Semantic Validity and Safety Bias.** Visual inspection (Figure 5) confirms semantic correctness. BPOP recovers the true fork–join structure on SRASEARCH and most true dependencies on the more complex EPIGENOMICS workflow despite multiple synchronization barriers. By enforcing global DAG constraints, Bayesian structure learning avoids the cyclic or infeasible graphs produced by local heuristics. Errors are safety-biased: on EPIGENOMICS, BPOP yields

more false positives (FP=20) than false negatives (FN=4). False positives correspond to conservative extra constraints that preserve safety at the cost of parallelism, whereas false negatives risk execution failure. Accordingly, BPOP compiles a risk-averse SOP by treating high-confidence edges as hard constraints and lower-confidence edges as advisory.

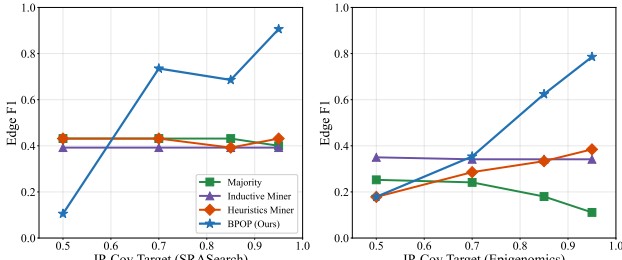

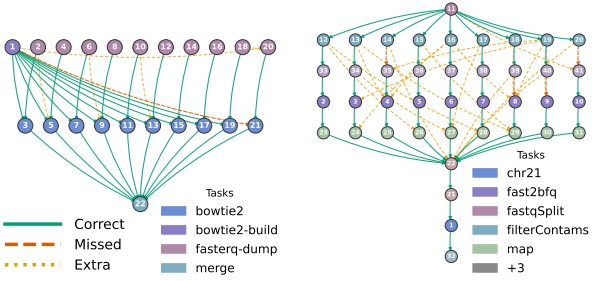

*Figure 5.* **Recovered vs. True SOPs (IP-Cov=0.95).** Green: correct dependencies; red: missed edges (unsafe); orange: false positives (safe). On SRASEARCH(Left), BPOP recovers the fork–join structure with F1=0.91 (TP=29/30, FN=1). On EPIGE-NOMICS(Right), BPOP recovers most true dependencies despite multiple synchronization barriers (F1=0.79, TP=44/48, FN=4)

*Figure 6.* **Structural Recovery vs. Trace Diversity (IP-Cov).** Edge-F1 as a function of trace diversity for SRASEARCH (left) and EPIGENOMICS (right). BPOP improves monotonically and outperforms baselines even at low diversity (IP-Cov $\approx$ 0.6), whereas baselines require near-complete ordering observations.

**Effect of Trace Diversity.** Figure 6 shows that BPOP benefits directly from increased trace diversity (IP-Cov), improving monotonically as more concurrent orderings are observed. In contrast, trace-only and process-mining baselines show limited or unstable gains, and may degrade as conflicting pairwise evidence accumulates. In particular, Inductive Miner overgeneralizes highly concurrent logs by collapsing actions into a single parallel block, discarding internal DAG structure. Overall, trace diversity is necessary but not sufficient: exposing concurrency alone is insufficient without a global, constraint-aware model.

### 5.5. Cloud Agent Experiment Results

We run MCMC for $10^6$ iterations with 50% burn-in, requiring 5–6 hours on 8 parallel workers and $\leq 500$ MB memory per scenario (See Table 15). We sweep the noise parameter $\varepsilon \in \{0.001, 0.005, 0.01, 0.05\}$ and trace diversity $IP\text{-}Cov \in \{0.6, 0.7, 0.8, 0.9, 1.0\}$. POs are estimated using the threshold estimator $\hat{h}_{\text{thr}}(\alpha)$ at the default $\alpha \approx 1/3$; see Appendix F.3.6 for threshold sensitivity analyses.

Figure 1 illustrates a representative Cloud-IaC workflow from this experiment, linking observed trace variation to the inferred partial order and compiled graph executor.

#### 5.5.1. STRUCTURAL RECOVERABILITY RESULTS

**Aggregate Performance at High Informativeness.** Figure 7 summarizes results at full trace diversity (IP-Cov= 1.0, $\epsilon = 0.01$). BPOP achieves the highest structural fidelity (Edge-F1 = 0.95), exceeding the best baseline (Heuristics

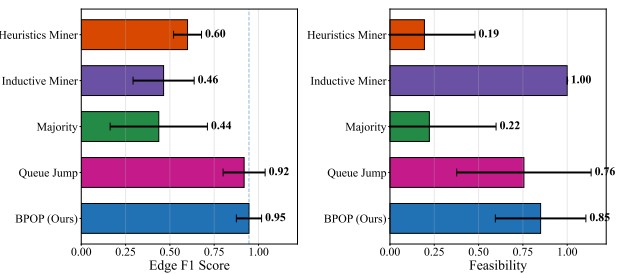

*Figure 7.* **Aggregate Performance (IP-Cov=1.0).** Edge-F1 (left) and feasibility (right) across cloud agents scenarios. BPOP achieves the best accuracy–validity trade-off, while baselines either overgeneralize or produce infeasible graphs.

Miner: 0.60). It is the only method that achieves *both* high structural accuracy *and* robust execution validity (0.85). Inductive Miner again attains perfect feasibility by learning overly permissive models with low precision, while Majority and Heuristics Miner often produce infeasible graphs.

**Effect of Trace Diversity.** As IP-Cov increases, BPOP improves monotonically, indicating that diverse traces exposing concurrency are critical for accurate recovery (Figure 8). In contrast, Majority and process-mining baselines show unstable behavior and may degrade as conflicting pairwise evidence accumulates. Bayesian Queue-Jump can be competitive at high IP-Cov but is less consistent and computationally impractical due to repeated NLE evaluations (Appendix F.2.4).

**Structural Fidelity vs. Execution Validity.** Baselines exhibit a clear accuracy–validity trade-off (Table 1). Inductive Miner achieves perfect feasibility by learning permissive models, while Heuristics Miner often over-constrains the graph and loses validity on harder workflows. BPOP combines strong structural recovery with high feasibility across IP-Cov settings, offering more practical accuracy–runtime trade-off than Bayesian QJ for larger workflows.

**Robustness to $\epsilon$.** At high trace diversity (IP-Cov $\geq 0.9$), BPOP's Edge F1 varies by less than 0.02 across the full

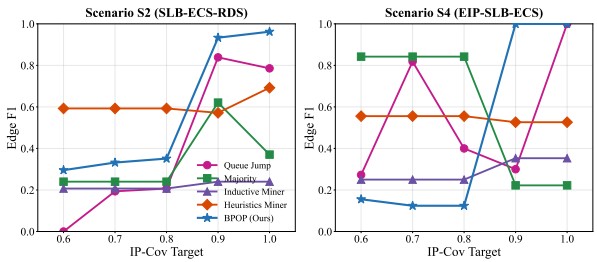

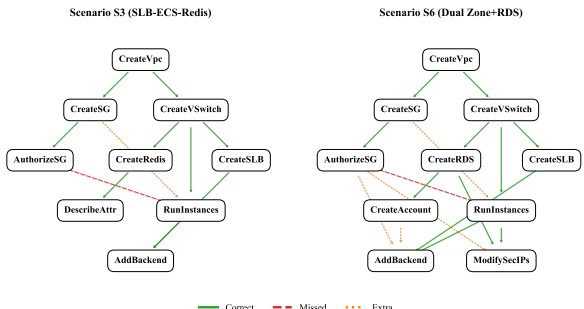

*Figure 8.* **Structural Recovery vs. Trace Diversity (Representative Scenarios).** Edge-F1 as a function of trace diversity (IP-Cov) for two cloud provisioning scenarios: S2 (SLB–ECS–RDS, left) and S4 (EIP–SLB–ECS, right). BPOP improves with increasing diversity and outperforms all baselines, while trace-only and process-mining methods exhibit unstable or degrading behavior.

*Table 1.* **Feasibility by IP-Coverage Target.** Fraction of observed traces that are valid linear extensions of the inferred partial order.

| Method | Target IP-Coverage | | | | |
| --- | --- | --- | --- | --- | --- |
| | 0.6 | 0.7 | 0.8 | 0.9 | 1.0 |
| Inductive Miner | 1.00 | 1.00 | 1.00 | 1.00 | 1.00 |
| Bayesian QJ | 0.54 | 0.60 | 0.69 | 0.71 | 0.76 |
| Majority | 0.67 | 0.67 | 0.50 | 0.22 | 0.22 |
| Heuristics Miner | 0.17 | 0.17 | 0.14 | 0.19 | 0.19 |
| **BPOP (Ours)** | **0.78** | **0.83** | **0.82** | **0.75** | **0.85** |

range of $\epsilon \in [0.005, 0.05]$. In contrast, varying IP-Cov from 0.6 to 1.0 (at fixed $\epsilon = 0.01$) changes F1 by 0.39—a 30× larger effect.(See Table 17 in Appendix).

To qualitatively validate these quantitative gains, Figure 9 visualizes the inferred SOPs against the ground truth.

### 5.5.2. EFFICIENT EXECUTION

We evaluate the 120-sweep experiment inference results from section 5.5.1 across 6 Cloud-IaC scenarios. See Appendix F.3.8 for an example user case comparing *Expert* and *Hybrid* modes on a concrete cloud provisioning task.

**Trace diversity drives a sharp transition from reactive to compiled execution.** Table 2 shows that increasing trace diversity (IP-Cov) yields a non-linear improvement in both structural recovery and execution. At low diversity (IP-Cov $\leq 0.8$), recovery remains limited (F1 $\leq 0.41$), leading to frequent fallback (33–50%) and high LLM overhead (4.0–5.9 calls/task; 17k–32k tokens/task), as the compiled plan under-specifies prerequisites. Once diversity reaches IP-Cov $\geq 0.9$, recovery becomes highly accurate, completeness reaches 100%, and fallback reasoning drops to zero. The remaining ~583 tokens/task come entirely from one-shot intent parsing rather than step-by-step reasoning. Scenario-level results (Table 18) further show that overhead is concentrated in complex multi-service workflows (`slb_ecs_rds`, `slb_ecs_redis`, and

*Figure 9.* **Qualitative Structure Recovery (IP-Cov=1.0).** Recovered SOPs for two representative Cloud-IaC scenarios: S3 (SLB–ECS–REDIS, left) and S6 (DUAL ZONE + RDS, right). Green edges denote correct dependencies, red denote missed edges (FN), and orange denote extra constraints (FP). BPOP recovers most true dependencies with safety-biased errors: S3 achieves TP=9, FP=1, FN=1; S6 achieves TP=11, FP=4, FN=1.

`eip_slb_ecs`), while simpler scenarios (`simple_ecs`, `dual_zone_ecs_slb`, and `dual_zone_ecs_slb_rds`) execute reliably once sufficient diversity is observed.

**Compiled POSET execution achieves both high success and efficiency.** Table 3 shows that Hybrid execution is the only mode that achieves *100% success* across all six scenarios, combining compiled POSET execution with limited fallback. While Expert execution is maximally efficient when correct (0 reasoning tokens; only ~583 intent-parsing tokens/task; 34.4 s total), this analysis excludes upfront costs: the token cost × NumTraces required to learn the structure. It fails on 2/6 scenarios due to the absence of recovery. Hybrid preserves most efficiency benefits of compilation while repairing these failures, requiring only 2 fallback events and 79k tokens in total. In contrast, pure LLM exploration is less reliable and far more expensive, with Explore modes consuming 234k–382k tokens and 1,319–2,580 s runtime.

**Qualitative graphs explain the cost collapse.** In Figure 9, missed edges (red) correspond to missing prerequisites (driving fallback), while false positives (orange) reduce parallelism but remain safe. At high IP-Cov, red edges are rare, consistent with 0% fallback and 0 reasoning tokens (~583 intent-parsing tokens/task remain as a fixed one-shot cost).

## 6. Related Work

**Structure Learning in Graphical Models.** Classical causal and graphical-model structure learning, such as PC (Spirtes et al., 2000) and NOTEARS (Zheng et al., 2018), aims to recover a DAG from i.i.d. samples under statistical assumptions on data. Bayesian-network structure learning has also used orders and partial orders as computational devices for searching or sampling over DAGs (Parviainen & Koivisto, 2010; Niinimäki et al., 2016; Kangas et al., 2016).

*Table 2.* **Effect of trace diversity (IP-Cov).** Higher IP-Cov improves structural recovery (Edge-F1) and reduces reliance on fallback planning. *LLM Calls/task* counts the number of pipeline decision stages that may invoke the LLM (the unit-cost intent-parsing stage is always counted as 1); *Tokens/task* reports total tokens consumed per task. [†]At IP-Cov $\geq$ 0.9, the $\sim$583 tokens/task come entirely from one-shot intent parsing; 0 reasoning tokens are generated.

| IP-Cov | F1 | Complete (%) | Fallback (%) | LLM Calls /task | Tokens /task |
|---|---|---|---|---|---|
| 0.6 | 0.329 | 70.8 | 50.0 | 5.9 | 32,208 |
| 0.7 | 0.350 | 58.3 | 50.0 | 5.6 | 29,359 |
| 0.8 | 0.413 | 75.0 | 33.3 | 4.0 | 17,479 |
| 0.9 | **0.872** | **100.0** | **0.0** | **1.0** | $\mathbf{\sim583}$[†] |
| 1.0 | 0.857 | **100.0** | **0.0** | **1.0** | $\mathbf{\sim583}$[†] |

*Table 3.* **Execution performance across six scenarios.** All modes use the partial order inferred at IP-Cov = 1.0. "Explore (No CoT)" removes explicit chain-of-thought but still uses the LLM for action selection. Totals and averages are over scenarios. *LLM tokens* counts total agent token use; Expert uses only one intent-parsing call and generates no reasoning tokens.

| Metric | Expert | Hybrid | Explore (No CoT) | Explore (CoT) |
|---|---|---|---|---|
| Success rate | 66.7% | 100.0% | 50.0% | 66.7% |
| Failures | 2 | 0 | 3 | 2 |
| Total actions | 44 | 57 | 41 | 53 |
| Total time (s) | 34.40 | 225.23 | 1318.69 | 2580.47 |
| LLM tokens | 3,501[‡] | 79,406 | 233,994 | 381,794 |
| Fallback count | – | 2 | – | – |
| Avg. time (s) | 5.73 | 37.54 | 219.78 | 430.08 |
| Avg. actions | 7.3 | 9.5 | 6.8 | 8.8 |
| Avg. tokens | $\sim$583[‡] | 13,234 | 38,999 | 63,632 |

These methods are closely related in spirit, but their observations are typically variable-valued samples rather than feasibility-constrained action traces. BPOP instead learns an executable precedence structure from sequential logs: the observed data are linearized executions, the likelihood respects which actions are feasible at each frontier.

**Partial Orders from Rankings and Mutation Traces.** Bayesian poset inference from rank data has been studied through random linear extensions (Nicholls et al., 2025) and Mallows-type noise models (Chuxuan et al., 2024). Order recovery from choice data also appears in settings such as top-$K$ recovery (Nguyen, 2022). A related biological line of work infers temporal constraints among accumulating genetic mutations, often using partial-order or conjunctive Bayesian-network models (Beerenwinkel et al., 2009; Gerstung et al., 2011). BPOP differs from these settings by observing complete action traces and by using the inferred order operationally: the goal is not only to estimate temporal precedence, but also to compile the posterior structure into a frontier-based executor.

**Planning from Traces.** Action model acquisition methods, such as ARMS (Yang et al., 2007) and FAMA (Aineto et al., 2019), reconstruct action schemata from traces, often using constraint-based or optimization-based formulations to handle partial observability. Recent work has also connected traces, event logs, and partial-order planning: Helal & Lakemeyer (2023) study partial-order plans for numeric tasks, while Park et al. (2024) use mined event-log behavior to guide planning structures. BPOP is complementary: rather than learning domain dynamics or guiding a planner, it infers a precedence poset from successful traces and compiles it into an uncertainty-aware frontier execution policy.

**Process Mining from Event Logs.** Process mining discovers workflow models from event logs, including classical approaches such as the $\alpha$-algorithm (Van der Aalst et al., 2004). These methods provide useful descriptive models of observed behavior, but can overfit incidental serializations and usually do not quantify uncertainty over latent precedence relations. BPOP instead targets a normative dependency structure: it separates necessary constraints from incidental ordering, quantifies recoverability through IP-Cov, aligns the graph with downstream frontier execution.

# 7. Conclusion

BPOP targets bounded, finite-horizon workflows and learns an executable precedence structure from successful traces. By distilling invariant dependency structure rather than memorizing linear scripts, it reduces redundant agent inference while preserving safety through uncertainty-aware compilation. Compared to process-mining baselines that prioritize fast discovery, BPOP trades offline compilation speed for principled uncertainty quantification and superior structural fidelity. This one-time inference cost is amortized across executions, enabling highly efficient, low-latency agent behavior at runtime. More broadly, BPOP complements agentic memory systems (e.g., LEGOMem (Han et al., 2025)) by generalizing across executions through explicit dependency structure rather than fixed action sequences.

# Impact Statement

This work uses execution traces to infer a partial-order dependency structure and compile it into a frontier-based execution policy, reducing repeated agent deliberation and inference/token cost by exposing parallelism and making constraints explicit and auditable. Risks include misuse in high-stakes automation or failure under distribution shift (e.g., changing tools or control-flow semantics). We mitigate these risks by scoping to bounded procedural settings, reporting recoverability diagnostics (IP-Cov) and calibrated uncertainty, and enabling conservative execution via confidence thresholding and fallbacks with human oversight.

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

# A. Preliminaries: Partial Orders

We follow standard terminology for partial orders; see, e.g., Brightwell (1993).

## A.1. Choice sets

Let $\mathcal{A}$ denote the universe of actions with $|\mathcal{A}| = m$. A **choice action set** $S$ is any non-empty subset of $\mathcal{M}$. We write

$$\mathcal{B}_{\mathcal{A}} \coloneqq \{S \subseteq \mathcal{A} : S \neq \emptyset\}.$$

In our setting, each observed trace is associated with a choice set of actions that were available/relevant for that execution instance.

## A.2. Strict partial orders and representations

**Definition A.1** (Strict partial order / poset). A **(strict) partially ordered set** (poset) is a pair $h = (X, \succ_h)$, where $X$ is a finite set and $\succ_h$ is a binary relation on $X$ that is: (i) *irreflexive* ($x \not\succ_h x$ for all $x$), (ii) *transitive* ($x \succ_h y$ and $y \succ_h z$ imply $x \succ_h z$).

Throughout this paper we take $X = \mathcal{A}$, and we index items by integers $\{1, \dots, m\}$ when convenient. We represent a strict partial order $h$ by a binary matrix $h \in \{0, 1\}^{m \times m}$ with

$$h_{ij} = 1 \iff i \succ_h j, \qquad h_{ii} = 0.$$

An illustrative example is

$$h = \begin{bmatrix} 0 & 0 & 1 & 0 & 1 \\ 0 & 0 & 1 & 1 & 1 \\ 0 & 0 & 0 & 0 & 1 \\ 0 & 0 & 0 & 0 & 0 \\ 0 & 0 & 0 & 0 & 0 \end{bmatrix}.$$

Two distinct items $i, j$ are **comparable** if either $i \succ_h j$ or $j \succ_h i$. They are **incomparable** otherwise, i.e.,

$$i \parallel_h j \iff h_{ij} = 0 \text{ and } h_{ji} = 0.$$

A strict order is **total** (linear) if every pair is comparable; it is **empty** (discrete) if $h_{ij} = 0$ for all $i \neq j$. A total order $\ell = (X, \succ_\ell)$ on a set $X = (1, 2, \dots, m)$ can equivalently be represented as a simple ordered list, so we sometimes abuse notation and treat total orders as if they were ordered lists $\ell = (\ell_1, \dots, \ell_m)$ satisfying $1 \leq i < j \leq m \iff \ell_i \succ_\ell \ell_j$.

**DAG view, closure, and cover.** A strict partial order corresponds to a directed acyclic graph (DAG) on vertex set $\mathcal{A}$, with an edge $i \to j$ whenever $i \succ_h j$. When $h$ contains *all* implied precedences (i.e., it is transitively closed), we denote it by $h$ (or explicitly $h^+$). For visualization and evaluation we often use the **cover relation** (Hasse diagram), obtained by the **transitive reduction** of $h$: it removes edges implied by transitivity while preserving reachability (and hence identifies the same partial order).

## A.3. Linear extensions

A **linear extension** of a poset $h = (X, \succ_h)$ is a total order $\ell$ on $X$ that is consistent with $\succ_h$:

$$i \succ_h j \implies i \text{ appears before } j \text{ in } \ell.$$

Given a trace $y = (y_1, \dots, y_T)$ containing a subset of actions, we say $y$ is consistent with $h$ if it does not violate any precedence constraints restricted to its realized items. Equivalently, for any $(i, j)$ with $i \succ_h j$ and both $i, j$ appearing in the trace, $i$ must appear before $j$ in $y$.

**Height (depth).** The **height** of a poset, denoted $\mathrm{ht}(h)$, is the length of a longest chain. For a total order on $m$ elements, $\mathrm{ht}(h) = m$; for the empty order, $\mathrm{ht}(h) = 1$.

## A.4. Partial Order Dimension

**Dimension and realizers.** The **dimension** of a poset measures how many total orders are required to represent it as an intersection.

**Definition A.2** (Dimension). Let $h = (X, \succ_h)$ be a poset on a finite set $X$. The *dimension* of $h$, denoted $\dim(h)$, is the smallest integer $K$ such that there exist linear extensions $\ell_1, \ldots, \ell_K$ satisfying

$$x \succ_h y \iff \big(x \text{ appears before } y \text{ in } \ell_k \text{ for every } k = 1, \ldots, K \big).$$

Equivalently,

$$h = \bigcap_{k=1}^{K} \ell_k.$$

**Definition A.3** (Realizer). A *realizer* of size $K$ for a poset $h = (X, \succ_h)$ is a family of $K$ linear extensions $\{\ell_1, \ldots, \ell_K\}$ whose intersection equals $h$. Thus, $\dim(h)$ is the size of the smallest realizer.

**Geometric view.** A classical interpretation due to Dushnik & Miller (1941) is that $\dim(h) \leq K$ iff the elements can be embedded in $\mathbb{R}^K$ such that $x \succ_h y$ corresponds to coordinate-wise dominance.

**Basic bounds and computational difficulty.** Dimension is bounded above by Hiraguchi's inequality (Hiraguchi, 1951; Bogart, 1973): for $m \geq 4$, $\dim(h) \leq \lfloor m/2 \rfloor$, and this is tight for the standard example (poset "crown") family. Computing $\dim(h)$ is NP-hard in general; Yannakakis (1982) establishes strong hardness results even for restricted families. This computational difficulty motivates approaches (including Bayesian ones) that infer plausible ranges of $K$ rather than computing $\dim(h)$ exactly.

## A.5. Counting Linear Extensions

**#P-hardness.** Counting linear extensions is computationally intractable in general: given a poset $h$, the quantity

$$L(h) = |\{\text{linear extensions of } h\}|$$

is #P-complete to compute exactly (Brightwell & Winkler, 1991). Consequently, likelihoods that require summing over all linear extensions (or exactly evaluating $L(h)$) are only feasible for small instances or special poset families.

**Exact counting via dynamic programming over ideals.** A classical exact strategy uses recursion over maximal elements:

$$L(h) = \sum_{x \in \max(h)} L(h \setminus \{x\}),$$

and memoizes subproblems over valid subsets (ideals/filters). This yields worst-case complexity $O(2^n n)$ but can be tractable for bounded-width or structurally sparse instances (see, e.g., (De Loof et al., 2006)).

**Exploiting structure (sparsity / decomposition).** Modern exact methods improve practical performance by decomposing subproblems into connected components and applying dynamic programming over poset ideals. These approaches can be highly effective for moderate $n$ when the underlying posets are sparse.

**Approximation and sampling.** For larger instances, practical toolchains rely on approximation and sampling-based estimators of $L(h)$ (e.g., (Talvitie & Koivisto, 2024)). These approximation routes motivate modeling choices that avoid exact counting inside the likelihood whenever possible.

# B. MCMC Implementation Details

We employ a Metropolis-Hastings-within-Gibbs sampler to infer the latent parameters. Table 4 summarizes the proposal distributions and acceptance criteria for each parameter block.

The sampler operates on a **randomized cycling scheme with dimension-proportional weights**. We assign update frequencies to each parameter block $i$ proportional to its weighted complexity. To prevent sequential bias and ensure ergodic mixing, we construct a discrete schedule list based on these weights within each cycle (length $L = 500$). This ensures high-dimensional parameters (e.g., latent utilities $U$) are updated frequently without introducing order-dependent correlations. Traces are stored every 100 iterations (thinning) to reduce autocorrelation.

*Table 4.* Summary of MCMC Transition Kernels.

| Parameter | Prior $p(\cdot)$ | Proposal $q(\cdot|\cdot)$ | Acceptance Ratio $\alpha$ | Weight |
|---|---|---|---|---|
| **Latent Utilities** $(U)$ | $U_i \sim \mathcal{N}(0, \Sigma_\rho)$ (equicorrelated) | **Random Walk(RW):** $U_i^* \sim \mathcal{N}(U_i, \Sigma_\rho)$ | $\frac{p(U^*)p(y|U^*)}{p(U)p(y|U)}$ | $N$ |
| **Correlation** $(\rho)$ | $\rho \sim \mathrm{Beta}(1, \alpha_\rho)$ | **Multiplicative RW:** $\rho^* = 1 - (1-\rho)\delta$ $\delta \sim \mathcal{U}(d_r, 1/d_r)$ | $\frac{p(\rho^*)p(U|\rho^*)}{p(\rho)p(U|\rho)} \cdot \frac{1}{\delta}$ | $2$ |
| **Softmax Temp** $(\beta)$ | $\beta \sim \mathrm{Gamma}(a, b)$ | **Log-Normal RW:** $\log \beta^* = \log \beta + \epsilon$ $\epsilon \sim \mathcal{N}(0, \sigma^2)$ | $\frac{p(\beta^*)p(y|\beta^*)}{p(\beta)p(y|\beta)} \cdot \frac{\beta^*}{\beta}$ | $2$ |
| **Dimension** $(K)$ | $K \sim \mathrm{Poisson}(\lambda)$ truncated at $K \geq 1$ | **Reversible Jump:** *Birth:* add column *Death:* delete column | $\frac{p(K^*)p(y|K^*)}{p(K)p(y|K)} \cdot \frac{q_\leftarrow}{q_\rightarrow}$ | $\max(3, N)$ |

## B.1. Appendix: Metropolis–Hastings update for inverse temperature $\beta$ under a Gamma prior

**Acceptance ratio (statement).** The MH log-acceptance ratio for the $\beta$ update (if $\mathcal{L}(\beta)$ is the log-likelihood evaluated at $\beta$, all other parameters fixed) is

$$\log \alpha_\beta = \Big[\mathcal{L}(\beta') - \mathcal{L}(\beta)\Big] + \Big[\log p(\beta') - \log p(\beta)\Big] + \log\left(\frac{\beta'}{\beta}\right). \tag{9}$$

The final term is the proposal-density ratio (equivalently a Jacobian term arising from proposing symmetrically in $\eta = \log \beta$).

**Proof of** (9)**.** The MH ratio is

$$\alpha_\beta = \min\left\{1, \frac{p(\mathcal{D} \mid \beta')p(\beta')}{p(\mathcal{D} \mid \beta)p(\beta)} \cdot \frac{q(\beta \mid \beta')}{q(\beta' \mid \beta)}\right\}.$$

Because $\eta' = \eta + \epsilon$ is a symmetric Gaussian random walk, the proposal is symmetric in $\eta$: $q_\eta(\eta' \mid \eta) = q_\eta(\eta \mid \eta')$. Transforming back to $\beta = \exp(\eta)$ yields (change of variables)

$$q(\beta' \mid \beta) = q_\eta(\eta' \mid \eta) \left|\frac{d\eta'}{d\beta'}\right| = q_\eta(\log \beta' \mid \log \beta) \frac{1}{\beta'}.$$

Similarly,

$$q(\beta \mid \beta') = q_\eta(\log \beta \mid \log \beta') \frac{1}{\beta}.$$

Since $q_\eta(\log \beta' \mid \log \beta) = q_\eta(\log \beta \mid \log \beta')$, the Gaussian terms cancel, and we obtain

$$\frac{q(\beta \mid \beta')}{q(\beta' \mid \beta)} = \frac{(1/\beta)}{(1/\beta')} = \frac{\beta'}{\beta}.$$

Taking logs and substituting into the MH ratio gives (9).

## B.2. Reversible–Jump Update of the Dimension K

To infer the latent dimensionality $K$, we employ a Reversible Jump MCMC (RJMCMC) scheme. We define the transition between dimensions $K$ and $K \pm 1$ using a birth-death process. In this setting we choose $\lambda = 3$ to express a preference for parsimony, effectively regularizing the model against overfitting.

**Prior on $K$.** We assume a Poisson($\lambda$) prior truncated to $K \geq 1$:

$$\pi(K) = \frac{e^{-\lambda}\lambda^K/K!}{1 - e^{-\lambda}}, \quad K = 1, 2, \dots$$

**Move Probabilities.** Let $\rho_{K,K'}$ denote the probability of proposing a move from $K$ to $K'$. We define:

$$\rho_{K,K+1} = \begin{cases} 1 & \text{if } K = 1, \\ 0.5 & \text{if } K \geq 2, \end{cases} \qquad \rho_{K,K-1} = \begin{cases} 0 & \text{if } K = 1, \\ 0.5 & \text{if } K \geq 2. \end{cases}$$

At each step, we draw $r \sim \text{Unif}(0,1)$. If $r < \rho_{K,K+1}$, we propose an *up* move ($K \to K+1$); otherwise, we propose a *down* move ($K \to K-1$).

### B.2.1. UP MOVE: $K' = K + 1$

We propose adding a new latent feature column at a random position $c$. To ensure high acceptance rates, we draw the new column values from their *conditional prior* given the existing columns.

1. **Choose insertion slot.** Sample $c \sim \text{Unif}\{0, \ldots, K\}$. The existing columns $d \geq c$ are shifted to $d+1$.

2. **Sample the new column conditionally.** For each item $j$, we draw the new value $U_{j,c}$ based on the correlation with the existing $K$ columns:
$$U_{j,c} \sim \mathcal{N}(\mu_j, \sigma_{\text{cond}}^2),$$
where
$$\mu_j = \frac{\rho}{1 + (K-1)\rho} \sum_{d=1}^{K} U_{j,d}, \qquad \sigma_{\text{cond}}^2 = \frac{1 + (K-1)\rho - K\rho^2}{1 + (K-1)\rho}.$$

**Metropolis–Hastings Ratio.** The acceptance ratio $R$ is given by:

$$R = \frac{\pi(U', K+1)}{\pi(U, K)} \times \frac{q(U, K \mid U', K+1)}{q(U', K+1 \mid U, K)} \times |J|.$$

The Jacobian $|J| = 1$ because the dimension change is a direct insertion without scaling. The posterior factorizes as $\pi(U', K+1) = \pi_{\text{new}}(U' \mid U, K) \cdot \pi(U \mid K) \cdot \pi(K+1)$. Crucially, because we propose $U'$ from the conditional prior, the term $\pi_{\text{new}}$ in the numerator exactly cancels the proposal density $q_{\text{new}}$ in the denominator. The old block $\pi(U \mid K)$ also cancels.

Thus, the ratio simplifies to the likelihood ratio times the prior and proposal move probabilities:

$$R = \frac{\pi(K+1)}{\pi(K)} \frac{\rho_{K+1,K}}{\rho_{K,K+1}} \frac{p_S(Y \mid h(U', \beta))}{p_S(Y \mid h(U, \beta))}.$$

Taking the logarithm:

$$\boxed{\begin{aligned} \log \alpha_+ \;=\;& \log \pi(K+1) - \log \pi(K) \\ & + \log p_S(Y \mid h(U', \beta)) - \log p_S(Y \mid h(U, \beta)) \\ & + \log \frac{\rho_{K+1,K}}{\rho_{K,K+1}}. \end{aligned}} \tag{10}$$

Note: For $K \geq 2$, the proposal ratio $\frac{\rho_{K+1,K}}{\rho_{K,K+1}} = 1$. For $K = 1$, it is 2.

### B.2.2. DOWN MOVE: $K' = K - 1$

1. **Choose deletion slot.** Sample $c \sim \text{Unif}\{0, \ldots, K-1\}$.

2. **Remove column $c$.** Construct $U'$ by deleting the $c$-th column from all latent matrices.

**Proposal Density.** The reverse proposal (which would re-insert the deleted column from the conditional prior) dictates the ratio. The down-move proposal density is simply:

$$q(U', K-1 \mid U, K) = \rho_{K,K-1} \cdot \frac{1}{K}.$$

**Metropolis–Hastings Ratio.** By symmetry with the Up move, the Gaussian terms for the deleted column in the numerator (current state posterior) cancel with the hypothetical reverse proposal density in the denominator. The acceptance probability becomes:

$$
\begin{aligned}
\log \alpha_- \;=\; & \log \pi(K-1) - \log \pi(K) \\
& + \log p_S(Y \mid h(U', \beta)) - \log p_S(Y \mid h(U, \beta)) \\
& + \log \frac{\rho_{K-1,K}}{\rho_{K,K-1}}.
\end{aligned}
\tag{11}
$$

For $K > 2$, the proposal ratio log-term is 0. For $K = 2$, the down move is always allowed (ratio term $\log(1/0.5) = \log 2$), and for $K = 1$, the down move is forbidden.

## C. Likelihood Details

### C.1. Frontier-Softmax Likelihood with Successor Utility

**Sequential Frontier Choice.** We model the generative process of a trace $y = (y_1, \ldots, y_T)$ as a sequential selection from the set of currently available actions. Let $R_t = \{y_t, \ldots, y_T\}$ denote the set of remaining actions at step $t$. Given a latent partial order $h$, the **Frontier** $\mathcal{F}_t(h)$ is the set of actions whose precedence constraints are fully satisfied:

$$
\mathcal{F}_t(h) = \{a \in R_t : \nexists b \in R_t \text{ s.t. } b \succ_h a\}.
$$

**Successor Utility** To differentiate between valid actions, we assume the agent is *rational*: it prefers actions that unlock the most future work (minimizing the makespan). We define the **Successor Score** $S_t(a)$ as the count of remaining actions strictly dependent on $a$:

$$
S_t(a) \triangleq |\{\, b \in R_t \setminus \{a\} : a \succ_h b \,\}| .
$$

We map this count to a utility score $Q_{\text{succ}}$ using a log-diminishing return function:

$$
Q_{\text{succ}}(a; h, t) = \begin{cases} \log(1 + S_t(a)) & \text{if } a \in \mathcal{F}_t(h), \\ -\infty & \text{if } a \notin \mathcal{F}_t(h). \end{cases}
\tag{12}
$$

The case $Q = -\infty$ enforces strict structural consistency (zero probability for invalid actions).

**Boltzmann Likelihood.** The probability of selecting action $y_t$ at step $t$ is modeled as a Boltzmann-rational policy over the frontier:

$$
p(y_t \mid y_{1:t-1}, h, \beta) = \frac{\exp(\beta \cdot Q_{\text{succ}}(y_t; h, t))}{\sum_{a' \in \mathcal{F}_t(h)} \exp(\beta \cdot Q_{\text{succ}}(a'; h, t))}.
\tag{13}
$$

The total log-likelihood of the trace $y$ is the sum of log-probabilities over $t = 1 \ldots T$.

**Theoretical Properties.** This formulation provides three key advantages for structure learning:

1. **Strict Structural Consistency:** If the observed action $y_t$ violates the partial order (i.e., $y_t \notin \mathcal{F}_t(h)$), then $Q_{\text{succ}}(y_t) = -\infty$ and the likelihood drops to zero. This ensures that the learned graph $h$ must be compatible with the observed topological order.

2. **Efficiency Bias (Topological Guidance):** Among topologically valid actions, the model does not treat them uniformly. The utility $Q_{\text{succ}}$ biases the likelihood towards graphs where the observed trace follows a "path strategy" (executing high-dependency nodes first). This aligns the learned structure with the rational intent of the agent, rather than just random valid permutations.

3. **Polynomial Tractability:** Unlike exact marginalization over all linear extensions (which is #P-complete), computing the frontier and successor counts is polynomial. The likelihood evaluates in $\mathcal{O}(T \cdot |\mathcal{A}|)$, scaling efficiently to long execution logs.

**Computational Complexity.** We analyze the cost of evaluating the trace likelihood $\log p(y \mid h, \beta)$ for a single trace of length $T$. Let $|\mathcal{A}|$ be the action space size and $|\succ_h|$ be the number of edges in the candidate poset $h$. Assuming the graph structure and successor counts $S(a)$ are pre-computed for the candidate $h$ (a one-time cost per MCMC step), the trace evaluation involves two operations at each step $t$:

1. **Frontier Maintenance:** We maintain the set of feasible actions $\mathcal{F}_t(h)$ using Kahn's algorithm logic (Kahn, 1962). Upon observing action $y_t$, we decrement the unmet-prerequisite counts for its children. Since each dependency edge $(u, v) \in \succ_h$ is processed exactly once over the full trace, the total maintenance cost is linear in the graph size: $\mathcal{O}(|\succ_h|)$.

2. **Policy Evaluation:** Computing the normalization constant for Eq. 13 requires summing the exponential utilities over the current frontier. With pre-computed successor scores, looking up $Q_{\text{succ}}(a)$ is $\mathcal{O}(1)$. The cost is thus proportional to the frontier size at each step: $\mathcal{O}(\sum_{t=1}^{T} |\mathcal{F}_t(h)|)$.

Combining these terms, and bounding the frontier size by $|\mathcal{A}|$, the total complexity per trace is:

$$\mathcal{C}_{\text{trace}} \in \mathcal{O}\left(|\succ_h| + \sum_{t=1}^{T} |\mathcal{F}_t(h)|\right) \subseteq \mathcal{O}(|\succ_h| + T \cdot |\mathcal{A}|). \tag{14}$$

This linear scaling in both graph density and trace length ensures the likelihood remains tractable for long execution logs, avoiding the factorial complexity of summing over all linear extensions.

### C.2. Likelihood

**Robust Mixture Model.** Real-world execution logs contain noise (e.g., asynchronous logging latency or manual interventions) that may appear to violate strict causal dependencies. To prevent the likelihood from collapsing to zero on these "trembling hand" errors, we define the choice probability as a mixture of a rational Boltzmann policy and a uniform noise distribution (Eq. 5).

- **Rational Component** $(1 - \epsilon)$**:** The agent selects $y_t \in \mathcal{F}_t(h)$ proportional to $\exp(\beta Q(y_t))$. If $y_t \notin \mathcal{F}_t(h)$ (a structural violation), this term is strictly zero.

- **Noise Component** $(\epsilon)$**:** The agent selects $y_t$ uniformly from all remaining actions $R_t$, ensuring a non-zero "safety floor" probability $\frac{\epsilon}{|R_t|}$ for any physically possible action.

This formulation allows BPOP to learn structure from noisy data: the gradient is driven by the rational component (maximizing topological fit), while the noise component acts as a robust buffer against outliers (See Figure 10 and Table 5 for the illustrated example).

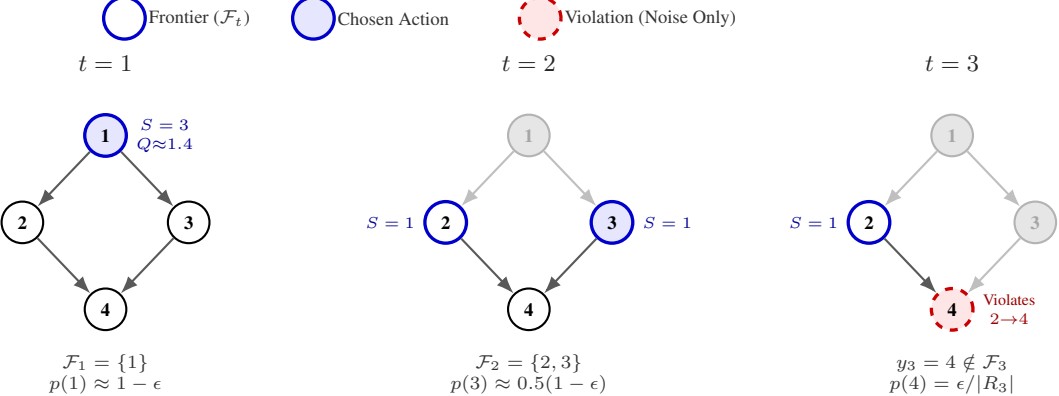

*Figure 10.* **Stepwise likelihood under the Robust Frontier-Softmax model.** We score actions by Successor Utility $S_t(a)$. (**Left**) $t = 1$: Action 1 is the only valid choice. (**Center**) $t = 2$: Actions 2 and 3 are symmetric choices. (**Right**) $t = 3$: Attempting action 4 (before 2) is a violation. The softmax term becomes 0, leaving only the "trembling hand" noise probability $\epsilon/|R_t|$.

## D. From Structure to Efficient Execution

### D.1. Trace Parsing and Action Definition

Let the raw agent session be a sequence of tokens $S = (o_1, \ldots, o_N)$ drawn from a mixed vocabulary $\mathcal{V} = \mathcal{A} \cup \mathcal{T}_{think}$.

*Table 5.* **Likelihood calculation for the violation trace** $y = (1, 3, 4, 2)$**.** The probability is a mixture of a rational softmax policy (over the frontier) and a uniform noise component. For the violation at $t = 3$, the rational component is zero, so the likelihood relies entirely on the noise floor.

| Step | State $(R_t, \mathcal{F}_t)$ | Utilities $S_t(\mathcal{F}_t)$ | Choice | Mixture Probability $p(y_t \mid \dots)$ |
|---|---|---|---|---|
| $t = 1$ | $R = \{1, 2, 3, 4\}$ | $S(1) = 3$ | **1** | $(1 - \epsilon) \cdot \underbrace{1.0}_{\text{Softmax}} + \underbrace{\frac{\epsilon}{4}}_{\text{Noise}}$ |
| | $\mathcal{F} = \{1\}$ | $Q \approx 1.39$ | | |
| $t = 2$ | $R = \{2, 3, 4\}$ | $S(2) = 1, S(3) = 1$ | **3** | $(1 - \epsilon) \cdot \underbrace{0.5}_{\text{Softmax}} + \frac{\epsilon}{3}$ |
| | $\mathcal{F} = \{2, 3\}$ | $Q \approx 0.69$ | | |
| $t = 3$ | $R = \{2, 4\}$ | $S(2) = 1$ | **4** | $\underbrace{0}_{\text{Invalid}} + \underbrace{\frac{\epsilon}{2}}_{\text{Safety Floor}}$ |
| | $\mathcal{F} = \{2\}$ | $Q(4) = -\infty$ | | *(Trembling Hand Only)* |
| $t = 4$ | $R = \{2\}$ | $S(2) = 0$ | **2** | $(1 - \epsilon) \cdot 1.0 + \epsilon$ |
| | $\mathcal{F} = \{2\}$ | $Q \approx 0.0$ | | |

- **Cognitive Space ($\mathcal{T}_{think}$):** Includes all tokens generated for planning, self-correction, or reflection (e.g., `Thinking:` `"I need to check the VPC ID..."`). These are treated as transient computational overhead.

- **Action Space ($\mathcal{A}$):** Includes only atomic, verifiable tool invocations that produce persistent side effects (e.g., `CreateInstance`, `blastn`). An action is typically a tuple $(f, \theta)$ of function identifier and arguments.

We define the training trace $y$ as the output of a projection operator $\Pi : \mathcal{V}^* \to \mathcal{A}^*$ that filters strictly for functional primitives:

$$y = \Pi(S) = (a_1, a_2, \dots, a_T) \quad \text{where } a_i \in S \cap \mathcal{A} \tag{15}$$

By discarding $\mathcal{T}_{think}$, BPOP effectively learns to compile the logic implicit in the reasoning steps directly into the structural dependencies of $y$.

**Actions as Expert-Polished Primitives.** Our definition of atomic actions $\mathcal{A}$ is grounded in the existence of **Standard Operating Procedures (SOPs)**. In high-stakes domains, human experts rely on "runbooks" or instruction booklets where each step has been carefully defined, polished, and validated to be safe and deterministic. For example, a cloud provider defines `CreateVPC` not as a vague intent, but as a precise contract with specific parameters and return values. In the "Enough Thinking" paradigm, we treat these actions as the fundamental units of truth. By projecting the agent's behavior onto this expert-defined subspace, we effectively align the agent's "muscle memory" with the polished instruction sets designed by system architects, discarding the noisy, ad-hoc reasoning that connects them.

### D.2. GEE Architecture: Decoupling Control Flow and Data Flow

The inferred SOP specifies control flow (what must precede what), but execution additionally requires data flow (how parameters propagate across tool calls).

**Execution State and Frontier Semantics** The executor maintains a completed set $S \subseteq \mathcal{A}$, per-action runtime status (pending/running/done/failed), and a global artifact store (blackboard) $\mathcal{B}$ for tool outputs. The set of currently feasible actions forms the *frontier*. Frontier semantics makes concurrency explicit: actions in $F(S; h)$ are not ordered by $h$ given $S$ and can be dispatched in parallel, subject to tool and rate constraints.

**IO registry and blackboard.** We attach an IO signature $\mathcal{R}(a) = (\mathcal{I}_a, \mathcal{O}_a)$ to each action, where $\mathcal{I}_a$ is the required input-slot set and $\mathcal{O}_a$ is the output-field set(See Table 6 as example). After executing $a$, GEE writes $\mathcal{O}_a$ to $\mathcal{B}$; before executing $b$, it fills $\mathcal{I}_b$ from $\mathcal{B}$. Missing inputs or API errors trigger a controlled fallback (Section D.3).

*Table 6.* A minimal IO registry example (cloud provisioning). Output fields (bold) are stored in the blackboard and can satisfy subsequent inputs.

| Action | Inputs | Outputs |
|---|---|---|
| CreateVpc | RegionId | **VpcId** |
| CreateVSwitch | **VpcId**, ZoneId | **VSwitchId** |
| RunInstances | **VSwitchId**, SecurityGroupId | **InstanceIds** |

## D.3. Tri-Modal Execution: Risk-Aware Automation Boundaries

Enterprise workflows require determinism and audit ability. BPOP provides three execution modes (See Figure 11) that trade off automation efficiency against failure handling, selected by the operator based on scenario maturity and risk tolerance. When the inferred SOP is stable and trusted, EXPERT mode delivers maximal efficiency with strict determinism; when additional resilience is desired, HYBRID mode adds automatic recovery; when bootstrapping a new scenario or exploring alternative paths, EXPLORE mode collects traces for future SOP inference.

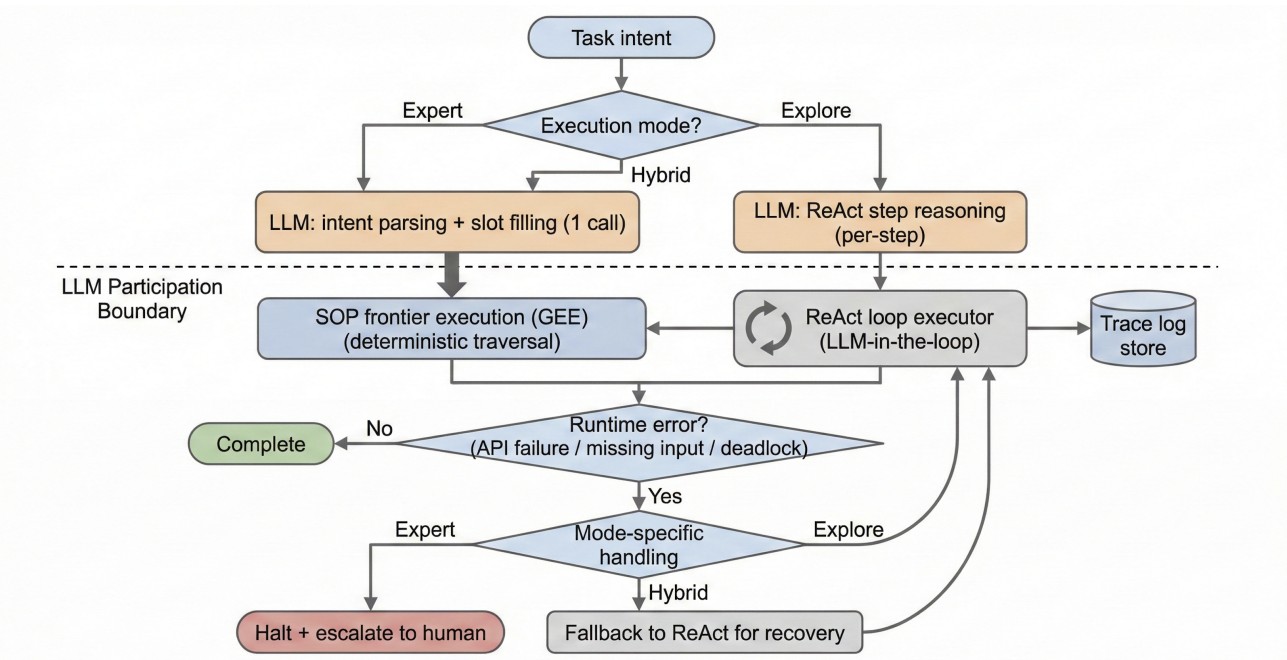

*Figure 11.* Tri-modal execution. EXPERT/HYBRID execute a posterior-compiled SOP via frontier scheduling; HYBRID falls back to an LLM planner on runtime errors; EXPLORE runs the full LLM loop to collect traces.

The three modes serve distinct operational scenarios:

- **EXPERT:** For stable, production-ready SOPs. The LLM performs a single intent-parsing and slot-filling call; execution is fully deterministic frontier traversal. On any error (API failure, missing input, deadlock), execution *halts immediately* and escalates to human operators. This provides maximal efficiency (∼1 LLM call) with strong stability and reproducibility.

- **HYBRID:** Designed for mature SOPs where resilience is desired. Execution follows the SOP identically to EXPERT, but upon error, it *falls back to step-by-step reasoning* for LLM-guided recovery instead of halting. When the SOP is correct, HYBRID matches EXPERT in efficiency; the difference lies purely in the error-handling strategy.

- **EXPLORE:** Targeted at cold-start scenarios (insufficient traces for SOP inference) or serving as the unconstrained baseline. The LLM operates in a full *reasoning-action loop* with complete execution history. By varying base models, temperature, and prompts, operators can explore diverse execution paths and collect traces for future SOP learning.

The key efficiency gain in EXPERT/HYBRID comes from *eliminating per-step LLM reasoning*—the inferred SOP encodes task structure, enabling deterministic frontier traversal rather than repeated LLM queries.

## E. Evaluation

### E.1. Incomparable-pair coverage and trace sufficiency

**Incomparable pairs.** Let $h^\star = (\mathcal{A}, \succ_{h^\star})$ be a strict partial order over $m = |\mathcal{A}|$ items. We define the set of *incomparable pairs* as distinct indices with no reachability in either direction:

$$\mathcal{P}_\|(h^\star) \;=\; \big\{(i,j) : 1 \le i < j \le m, \; i \nsucc_{h^\star} j \wedge j \nsucc_{h^\star} i\big\}.$$

**Incomparable-pair coverage (IP-Cov).** Each trace $y \in \mathcal{D}$ induces a total ordering over items, denoted $i \succ_y j$ if $i$ precedes $j$ in sequence $y$. We measure the diversity of the trace set $\mathcal{D}$ by quantifying how many ground-truth incomparable pairs are observed in *both* directions:

$$\text{IP-Cov}(\mathcal{D}; h^\star) \;=\; \frac{1}{|\mathcal{P}(h^\star)|} \sum_{(i,j)\in\mathcal{P}(h^\star)} \mathbb{I}\Big[\exists\, y, y' \in \mathcal{D} : (i \succ_y j) \wedge (j \succ_{y'} i)\Big].$$

Intuitively, IP-Cov reports the fraction of incomparable pairs that are statistically distinguishable from strict precedence given the observed data.

**Trace Sufficiency (Discussion).** We define a trace set $\mathcal{D}$ as *sufficient* for recovering $h^\star$ if $\text{IP-Cov}(\mathcal{D}; h^\star) = 1$. This condition guarantees that every ground-truth incomparable pair is observed in both relative orderings, providing the statistical evidence necessary to distinguish concurrency from causality. In practice, IP-Cov acts as a tractable surrogate for the theoretical ideal of observing all linear extensions, serving as a quantifiable control knob for dataset diversity in our recoverability experiments.

### E.2. Practical Trace Acquisition Strategy

In real-world deployments, the ground-truth partial order $h^\star$ is unknown, making the calculation of IP-Cov impossible. To approximate trace sufficiency and ensure the recovered SOP is not biased by a single planner's "habits," we employ a **Heterogeneous Model Exploration** strategy combined with a **Saturation-Based Stopping Criterion**.

**1. Heterogeneous Model Ensembling.** Standard LLMs exhibit distinct inductive biases in sequential planning. For example, given two concurrent tasks (e.g., `InitializeDB` and `ConfigNetwork`), Model A may deterministically prefer ordering $i \to j$, while Model B may prefer $j \to i$. Relying on a single model often leads to *false causality*—inferring a dependency where none exists.

To mitigate this, we generate the trace set $\mathcal{D}$ using an ensemble of distinct LLM backbones $\mathcal{M} = \{m_1, \ldots, m_k\}$ (e.g., Qwen-Plus, DeepSeek-V3.2, Kimi-K2, GLM-4.7). This diversity maximizes the entropy of the induced total orders:

$$\mathcal{D} = \bigcup_{m \in \mathcal{M}} \text{GenerateTraces}(m, \text{temperature} > 0.7).$$

By aggregating traces from diverse sources, we significantly increase the probability that true incomparable pairs are witnessed in opposing relative orders ($i \succ_y j$ and $j \succ_{y'} i$), allowing the intersection-based inference to correctly identify them as concurrent.

**2. Trace Diversity Saturation (Stopping Criterion).** Since we do not infer the graph during data collection, we monitor the raw traces for **pairwise saturation**. We track the set of item pairs observed in both relative directions (the "flipped" pairs):

$$\mathcal{R}_N = \{(i,j) \mid \exists\, y, y' \in \mathcal{D}_N : (i \text{ precedes } j \text{ in } y) \wedge (j \text{ precedes } i \text{ in } y')\}.$$

We stop collecting data when the size of $\mathcal{R}_N$ plateaus (i.e., $|\mathcal{R}_{N+\Delta}| \approx |\mathcal{R}_N|$). This indicates that adding more traces is no longer revealing new concurrency, suggesting that the pairs which have *never* flipped are likely true causal dependencies.

### E.3. Transitive Closure vs. Transitive Reduction

Given a partial order $h = (\mathcal{A}, \succ_h)$, we evaluate recovery performance on two levels:

**Transitive Closure (Semantics).** $\text{TC}(h)$ is the exhaustive set of all precedence pairs $(i, j)$ such that $i \succ_h j$. Metrics computed on TC assess whether the inferred order captures the correct causal flow, regardless of redundancy. This is the standard for checking logical consistency.

**Transitive Reduction (Skeleton).** $\text{TR}(h)$ is the minimal subset of dependencies required to induce $h$. It consists strictly of *covering pairs*: $(i, j)$ such that $i$ precedes $j$ with no intermediate action $k$ between them ($i \succ_h k \succ_h j$). Metrics computed on TR assess the ability to recover the clean, minimal "skeleton" of the workflow, which is critical for interpretability and efficient graph execution.

**Metric Selection Strategy.** We utilize $\text{TR}(\cdot)$ and $\text{TC}(\cdot)$ to address distinct evaluative questions:

- **Skeleton Recovery (Precision, Recall, F1, SHD):** We compare the inferred *covering relation* $\text{TR}(\widehat{h})$ against the ground truth $\text{TR}(h^\star)$. This evaluates whether we have recovered the *minimal executable SOP* without penalizing the omission of redundant transitive edges (which are logically implied but structurally unnecessary).

- **Feasibility (Consistency with Data):** Feasibility asks whether the inferred logic admits the observed traces. Since a valid trace must respect *all* implied precedence constraints, this property is defined with respect to the transitive closure:

$$\text{Feas}(\widehat{h}; \mathcal{D}) = \frac{1}{|\mathcal{D}|} \sum_{y \in \mathcal{D}} \mathbb{I}\left[ y \in \mathcal{L}(\widehat{h}) \right].$$

  Note that $y \in \mathcal{L}(\widehat{h})$ if and only if $y$ respects every constraint in $\text{TC}(\widehat{h})$.

- **IP-Cov (Trace Diversity):** The set of ground-truth incomparable pairs is defined by *mutual non-reachability*. Therefore, it must be computed from the zeros of the ground-truth closure $\text{TC}(h^\star)$:

$$\mathcal{P}_\parallel(h^\star) = \{(i, j) : i < j, \ (i, j) \notin \text{TC}(h^\star), \ (j, i) \notin \text{TC}(h^\star)\}.$$

  IP-Cov then measures the fraction of pairs in $\mathcal{P}_\parallel(h^\star)$ that are witnessed in both relative orders across the trace set.

### E.4. Efficiency Evaluation Metric Definition

Table 7 provides formal definitions for all metrics used in our evaluation.

*Table 7.* **Metric definitions.**

| Metric | Definition |
| --- | --- |
| Success Rate | Proportion of tasks completing without API errors |
| Completeness | Proportion of tasks executing all expert-required actions |
| Task Fallback | Proportion of tasks triggering $\geq 1$ LLM fallback |
| Action Fallback | Ratio of post-fallback actions to total actions |
| LLM Calls | Intent parsing (1) + Step by step reasoning steps |
| Tokens | Total input + output tokens consumed by LLM |
| Cover-F1 | F1 score of inferred vs. ground-truth cover edges |
| Fallback Layer | Poset layer index when fallback triggers |

## F. Experiment

### F.1. Baselines.

We compare against four baselines: (i) Majority, (ii) Inductive Miner (IMf), (iii) Heuristics Miner (HM), and (iv) Bayesian Queue-Jump (QJ). IMf and HM follow standard process-discovery pipelines from event logs (Leemans et al., 2013; Weijters et al., 2006). All baselines produce a directed acyclic *cover* graph by: (a) extracting a precedence graph, (b) greedily breaking cycles, and (c) projecting to a cover via transitive closure + transitive reduction.

#### F.1.1. Algorithm: Cycle Breaking

See Algorithm 1 for detail.

---

**Algorithm 1:** CYCLEBREAKANDCOVER$(A, W)$

---

**Require:** $A \in \{0, 1\}^{n \times n}$ adjacency; $W \in \mathbb{R}_{\geq 0}^{n \times n}$ weights (larger = stronger).

**Ensure:** DAG cover $\widehat{H}$.

1 **while** HASCYCLE$(A)$ **do**
2 $\quad$ $C \leftarrow$ FINDCYCLE$(A)$ $\qquad\qquad\qquad\qquad\qquad\qquad\qquad$ // any directed cycle
3 $\quad$ $(u, v) \leftarrow \arg\min_{(i,j) \in C} W_{ij}$ $\qquad\qquad\qquad\qquad\qquad$ // weakest edge
4 $\quad$ $A_{uv} \leftarrow 0$
5 $\widetilde{H} \leftarrow$ TRANSITIVECLOSURE$(A)$
6 $\widehat{H} \leftarrow$ TRANSITIVEREDUCTION$(\widetilde{H})$
7 **return** $\widehat{H}$

---

### F.1.2. BASELINE 1: MAJORITY

See Algorithm 2 for detail.

---

**Algorithm 2:** Majority baseline (pairwise precedence $\rightarrow$ cycle breaking)

---

**Require:** Orders $\mathcal{O} = \{o^{(t)}\}_{t=1}^{N}$ over items $[n]$; threshold $\tau$ (default 0.5).

**Ensure:** DAG cover $\widehat{H}_{\text{maj}}$.

1 $C \leftarrow 0_{n \times n}; \quad T \leftarrow 0_{n \times n}$
2 **for** $t \leftarrow 1$ **to** $N$ **do**
3 $\quad$ compute positions $\text{pos}^{(t)}(\cdot)$ in $o^{(t)}$
4 $\quad$ **foreach** $i \neq j \in o^{(t)}$ **do**
5 $\quad\quad$ $T_{ij} \leftarrow T_{ij} + 1$ $\qquad\qquad\qquad\qquad\qquad\qquad\qquad$ // pairs co-occurring
6 $\quad\quad$ **if** $\text{pos}^{(t)}(i) < \text{pos}^{(t)}(j)$ **then**
7 $\quad\quad\quad$ $C_{ij} \leftarrow C_{ij} + 1$

8 **foreach** $(i, j)$ *with* $T_{ij} > 0$ **do**
9 $\quad$ $p_{ij} \leftarrow C_{ij}/T_{ij}$
10 $A \leftarrow 0_{n \times n}; \quad W \leftarrow 0_{n \times n}$
11 **foreach** $\{i, j\}$ *with* $i < j$ **do**
12 $\quad$ **if** $p_{ij} > \tau$ **and** $p_{ij} > p_{ji}$ **then**
13 $\quad\quad$ $A_{ij} \leftarrow 1; \quad W_{ij} \leftarrow |p_{ij} - 0.5|$
14 $\quad$ **else if** $p_{ji} > \tau$ **and** $p_{ji} > p_{ij}$ **then**
15 $\quad\quad$ $A_{ji} \leftarrow 1; \quad W_{ji} \leftarrow |p_{ji} - 0.5|$
16 $\widehat{H}_{\text{maj}} \leftarrow$ CYCLEBREAKANDCOVER$(A, W)$
17 **return** $\widehat{H}_{\text{maj}}$

---

### F.1.3. BASELINE 2: INDUCTIVE MINER (IMF)

See Algorithm 3 for detail.

### F.1.4. BASELINE 3: HEURISTICS MINER

See Algorithm 4 for detail.

### F.1.5. BASELINE 4: BAYESIAN QUEUE JUMP

The Queue-Jump (QJ) baseline (Nicholls et al., 2025) models each observed trace $y = (y_1, \ldots, y_N)$ as a sequential choice process on a latent poset $H$. At step $j$, let $R_j$ be the set of remaining actions and $H[R_j]$ the induced subposet. The noise-free probability of selecting the next action is

$$q_{R_j}(y_j \mid H[R_j]) = \frac{\#\{\text{linear extensions of } H[R_j] \text{ that start with } y_j\}}{\#\{\text{linear extensions of } H[R_j]\}}.$$

---

**Algorithm 3:** Inductive Miner IMf (process discovery → footprints → precedence)

---

**Require:** Orders $\mathcal{O} = \{o^{(t)}\}_{t=1}^{N}$ over task names $V$; IMf noise threshold $\eta$.
**Ensure:** DAG cover $\widehat{H}_{\text{IMf}}$.

1  $L \leftarrow$ EVENTLOGFROMORDERS($\mathcal{O}$)                                   // each order = one case
2  $T \leftarrow$ INDUCTIVEMINERIMF($L; \eta$)                                        // process tree
3  $FP \leftarrow$ FOOTPRINTS($T$)                                    // fallback to log if needed
4  $Seq \leftarrow FP.\text{SEQUENCE}$                                              // ordered pairs $(a, b)$
5  $A \leftarrow 0_{|V| \times |V|}$;   $W \leftarrow 0_{|V| \times |V|}$
6  **foreach** $(a, b) \in Seq$ **do**
7  $\quad \lfloor\ A_{ab} \leftarrow 1$;   $W_{ab} \leftarrow 1$
8  $\widehat{H}_{\text{IMf}} \leftarrow$ CYCLEBREAKANDCOVER($A, W$)
9  **return** $\widehat{H}_{\text{IMf}}$

---

**Algorithm 4:** Heuristics Miner (dependency graph → precedence)

---

**Require:** Orders $\mathcal{O} = \{o^{(t)}\}_{t=1}^{N}$ over task names $V$; dependency threshold $\delta$.
**Ensure:** DAG cover $\widehat{H}_{\text{HM}}$.

1  $L \leftarrow$ EVENTLOGFROMORDERS($\mathcal{O}$)
2  $HN \leftarrow$ HEURISTICSMINER($L; \delta, \dots$)
3  $Dep \leftarrow HN.\text{DEPENDENCYMATRIX}$
4  $A \leftarrow 0_{|V| \times |V|}$;   $W \leftarrow 0_{|V| \times |V|}$
5  **foreach** $a \neq b \in V$ **do**
6  $\quad$ **if** $Dep(a, b) \geq \delta$ **then**
7  $\quad \quad \lfloor\ A_{ab} \leftarrow 1$;   $W_{ab} \leftarrow Dep(a, b)$
8  $\widehat{H}_{\text{HM}} \leftarrow$ CYCLEBREAKANDCOVER($A, W$)
9  **return** $\widehat{H}_{\text{HM}}$

---

To allow violations of feasibility, QJ mixes this with a "jump" distribution. In the plain QJ variant, the jump is uniform on remaining actions, $\pi_{\text{jump}}(y_j \mid R_j) = 1/|R_j|$, giving the stepwise likelihood

$$p(y_j \mid H[R_j], p) = (1 - p)\, q_{R_j}(y_j \mid H[R_j]) + p\, \pi_{\text{jump}}(y_j \mid R_j),$$

and the full-trace likelihood factors as $\prod_{j=1}^{N} p(y_j \mid H[R_j], p)$.

Evaluating $q_{R_j}(\cdot)$ requires counting linear extensions of $H[R_j]$ (and, for each candidate $y_j$, counting extensions conditioned to start with $y_j$). Counting linear extensions is #P-complete in general (Brightwell & Winkler, 1991), so QJ must repeatedly invoke an exponential-time subroutine across steps, traces, and MCMC iterations. This makes QJ impractical beyond small graphs; in our experiments it incurs hundreds of NLE calls per iteration and quickly becomes prohibitive as $|A|$ grows (Appendix F.2.4).

### F.2. WFCommons Experiment

#### F.2.1. WFCOMMONS WORKFLOW DATASET

**Source and format.** We use workflow execution instances from WFCommons WfInstances.Each workflow execution instance is represented as a WfFormat JSON file describing an actual execution on a distributed platform and includes (i) a workflow specification DAG (task dependencies) and (ii) time-stamped task execution information. Per-workflow dataset documentation is provided in the WfInstances application READMEs (WfCommons Project, 2021).

**Workflows used.** Figure 12 visualizes the ground-truth dependency structures (DAG covers) for the two selected benchmarks: 1. **SRASearch (Left):** A data-retrieval workflow (22 tasks) characterized by a "fork-join" pattern, where parallel download tasks eventually merge into a final analysis step. 2. **Epigenomics (Right):** A genomics pipeline (41 tasks) with high parallel width (independent branches) and multi-stage synchronization points, offering a more complex structural recovery challenge.

**Trace construction (observed linearizations).** For each WFCommons instance, we treat the specification DAG as ground truth and convert each execution log into an observed linearization by ordering tasks by their recorded start times. When two

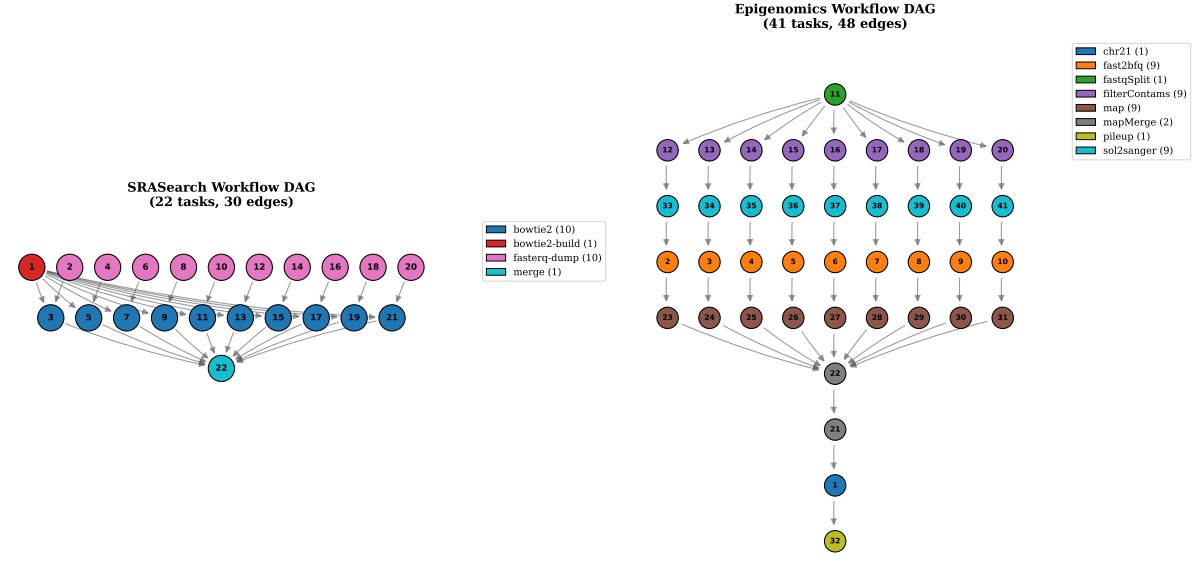

*Figure 12.* **Ground-Truth WFCommons Structures. (Left)** `SRASearch` (22 nodes, 30 edges) exhibits simple parallel-download logic. **(Right)** `Epigenomics` (41 nodes, 48 edges) features complex interleaved chains and synchronization barriers. BPOP aims to recover these topologies solely from linearized execution logs.

tasks share the same start time (or when timestamps are missing/identical after parsing), we apply a deterministic tie-break rule (lexicographic by task identifier) to produce a total order. Formally, for tasks $u, v$, we sort by the key

$$\kappa(u) \; = \; \big(\texttt{start\_time}(u), \; \texttt{task\_id}(u)\big),$$

and define the observed trace $\pi$ as the resulting ordered list of tasks.

**Ground truth target for recovery metrics.** Let $h$ be the workflow specification DAG with node $\mathcal{A}$. Because our model targets minimal precedence constraints, we evaluate recovery against the *cover graph* (also called the transitive reduction) $E_{\mathrm{cov}}$, which removes edges implied by transitivity. We report Cover-F1 and SHD computed on $E_{\mathrm{cov}}$, as well as feasibility and IP-Cov as described in the main text.

### F.2.2. SYNTHETIC IP-COVERAGE TARGETS

**Synthetic Trace Generation.** To systematically stress-test structural recovery under controlled diversity, we generate synthetic trace sets derived from the ground-truth DAGs of the SRASEARCH and EPIGENOMICS benchmarks. We model traces as linear extensions of the underlying partial order, sampled via a randomized Kahn's algorithm that selects uniformly from the available frontier at each step. To curate datasets with precise diversity levels, we employ an iterative greedy sampling procedure: starting from an empty set, we generate candidate linear extensions and retain only those that increase the *Incomparable Pair Coverage* (IP-Cov)—specifically, those that reveal a previously unobserved ordering direction for concurrent pairs—until a target coverage threshold $\tau \in \{0.50, 0.70, 0.85, 0.95\}$ is met( See table 8 for detail).

### F.2.3. COMPUTATIONAL COST DETAILS

Table 9 details the runtime for each of the 8 experimental configurations (1,000,000 MCMC iterations per run). The inference tasks for different IP-Cov targets and workflows are independent, we execute them in parallel on an 8-core instance. This reduces the effective wall-clock time to the duration of the longest single run ($\approx$ **4.6 hours**), making the approach feasible for overnight learning.

We discard the first $50\%$ ($500,000$ samples) as burn-in to ensure convergence to the stationary distribution.

*Table 8.* **Data Efficiency Analysis (WFCommons).** Number of execution traces required to reach specific IP-Coverage targets. `Epigenomics`, with a larger state space (576 incomparable pairs vs. 190 for `SRASearch`), requires nearly 2× more traces to achieve high diversity (0.95), illustrating the impact of topological complexity on data requirements.

| Workflow | Tasks ($|A|$) | Target IP-Cov | Realized IP-Cov | Num Traces |
|---|---|---|---|---|
| SRASEARCH | 22 | 0.50 | 0.689 | 7 |
| | | 0.70 | 0.758 | 8 |
| | | 0.85 | 0.868 | 9 |
| | | 0.95 | 0.963 | 10 |
| EPIGENOMICS | 41 | 0.50 | 0.521 | 3 |
| | | 0.70 | 0.738 | 4 |
| | | 0.85 | 0.863 | 8 |
| | | 0.95 | 0.951 | 19 |

*Table 9.* **MCMC Runtime on WFCommons Benchmarks.** Runtime is reported for 1M iterations on a standard CPU. By parallelizing the 8 independent experiments, the total turnaround time is determined by the single slowest run (275 min), rather than the sequential sum (∼20 hours).

| Workflow | IP-Coverage | | Traces ($N$) | Runtime (min) |
|---|---|---|---|---|
| | Target | Realized | | |
| SRASearch | 0.50 | 0.689 | 7 | 103.5 |
| | 0.70 | 0.758 | 8 | 96.9 |
| | 0.85 | 0.868 | 9 | 101.0 |
| | 0.95 | 0.963 | 10 | 114.0 |
| Epigenomics | 0.50 | 0.521 | 3 | 145.0 |
| | 0.70 | 0.738 | 4 | 143.5 |
| | 0.85 | 0.863 | 8 | 195.5 |
| | 0.95 | 0.951 | 19 | **275.3** |
| **Total (Sequential)** | | | | **1,174.7** |
| **Wall-Clock (8x Parallel)** | | | | **275.3** |

### F.2.4. QUEUE-JUMP (NLE) RUNTIME DIAGNOSTICS

Table 10 shows the resulting per-iteration cost on SRASEARCH when NLE is called repeatedly during MCMC. These measurements illustrate why the QJ baseline is not practical for larger WFCommons graphs.

*Table 10.* **Queue-Jump MCMC cost from runtime logs (SRASearch).** The baseline requires hundreds of NLE calls per iteration, leading to prohibitive runtimes.

| Setting | Value | Unit |
|---|---|---|
| Workflow size ($|V|$) | 22 | tasks |
| Num. traces | 16 | traces |
| NLE calls / iteration | 704 | calls |
| Time / iteration | 42.2 | seconds |
| Projected time (10k iters) | 117.3 | hours |
| Projected time (100k iters) | 1173.3 | hours |

### F.2.5. THRESHOLD SELECTION

The table F.2.5 details the threshold selection for inference. Individual topologies exhibit distinct preferences: the simpler SRASearch favors a conservative threshold ($\alpha = 0.5$) to ensure precision, whereas the highly parallel Epigenomics pipeline benefits from the theoretical baseline ($\alpha = 1/3$) to maximize recall of concurrent branches. These values were used for the qualitative DAG visualizations when we recover the true partial orders in Figure 12 in Appendix F.2.

### F.2.6. POSTERIOR DIAGNOSTICS (WFCOMMONS)

To validate inference stability, we examine the MCMC traces for both scientific workflows.

*Table 11.* **Impact of Topology on Threshold Sensitivity.** We observe a distinct divergence based on graph complexity: the simpler, structured `SRASearch` workflow benefits from conservative pruning ($\alpha = 0.50$), while the highly parallel `Epigenomics` pipeline requires permissive thresholds lower to preserve valid concurrent edges.

| | **SRASearch** (Simple) | | **Epigenomics** (Complex) | |
| --- | --- | --- | --- | --- |
| **Threshold** | **Edge F1** | **SHD $\downarrow$** | **Edge F1** | **SHD $\downarrow$** |
| $\alpha = 0.30$ | 0.841 | 11.0 | **0.811** | **21.0** |
| $\alpha = 1/3$ | – | – | 0.786 | 24.0 |
| $\alpha = 0.40$ | 0.879 | 8.0 | 0.737 | 30.0 |
| $\alpha = 0.50$ | **0.906** | **6.0** | 0.713 | 33.0 |

**SRASearch (Figure 13):** The sampler converges rapidly, estimating a low noise level ($\rho \approx 0.23$) and a topological depth of $K \approx 4.2$. This confirms the workflow is relatively clean and shallow.

**Epigenomics (Figure 14):** Reflecting its complex parallel structure, the model infers a higher noise parameter ($\rho \approx 0.55$) and a deeper topology ($K \approx 6.0$). Despite the higher complexity, the log-likelihood trace indicates stable mixing after burn-in.

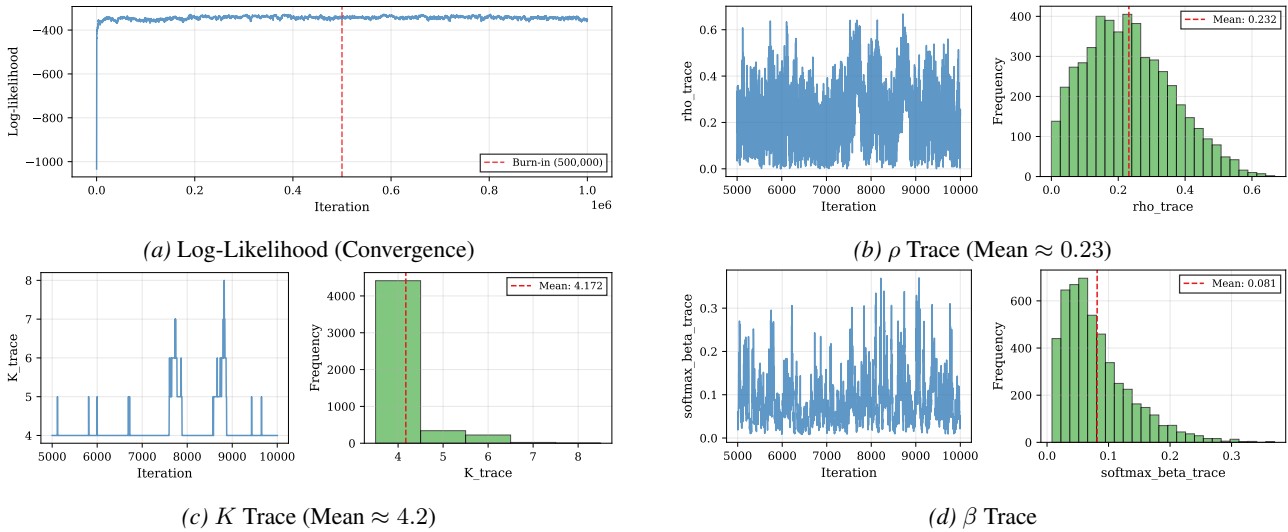

*(a)* Log-Likelihood (Convergence)

*(b)* $\rho$ Trace (Mean $\approx 0.23$)

*(c)* $K$ Trace (Mean $\approx 4.2$)

*(d)* $\beta$ Trace

*Figure 13.* **MCMC Diagnostics: SRASearch.** The traces show stable convergence to a low-noise, moderate-depth posterior.

## F.3. Cloud-IaC-6 Experiment

### F.3.1. CLOUD IAC 6 DATASET

We evaluate our method on **Cloud-IaC-6**, a benchmark of cloud provisioning tasks ranging from simple instance creation to complex high-availability clusters (see Table 14 and 12). Those scenarios are named from underlying Cloud Infrastructure product (See Table 13). The dataset contains 54 successful execution traces generated by a diverse pool of LLM agents (including Qwen-Plus, DeepSeek, and Kimi) to ensure behavioral diversity. The ground-truth graphs are provided by domain experts, see Figure 16. We provide the full implementation and benchmark datasets in our public repository.[2]

---

[2]`https://github.com/hollyli-dq/po_inference_agent/blob/main/data/cloud_iac_dataset/README.md`

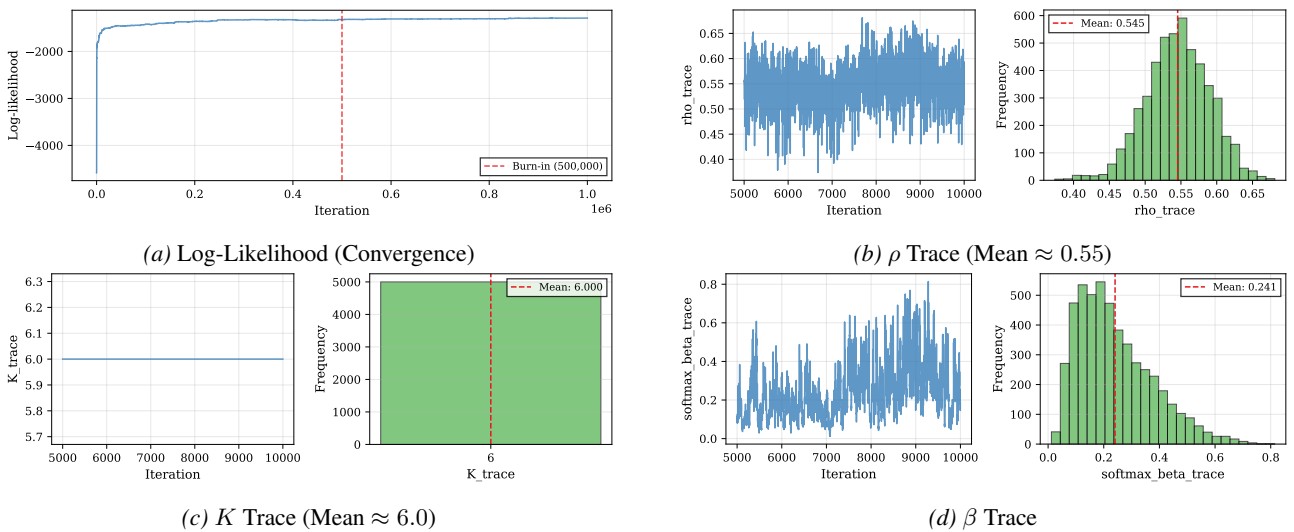

*(a)* Log-Likelihood (Convergence)

*(b)* $\rho$ Trace (Mean $\approx 0.55$)

*(c)* $K$ Trace (Mean $\approx 6.0$)

*(d)* $\beta$ Trace

*Figure 14.* **MCMC Diagnostics: Epigenomics.** The sampler stabilizes at a higher depth ($K \approx 6$) and noise level ($\rho \approx 0.55$), consistent with the workflow's complexity.

| ID | Scenario Identifier | Description |
|----|---------------------|-------------|
| 1 | SIMPLE_ECS | Provisions a VPC, VSwitch, and Security Group, followed by a single ECS instance. |
| 2 | SLB_ECS_RDS | A classic 3-tier web architecture integrating Server Load Balancer (SLB), ECS, and Relational Database Service (RDS). |
| 3 | SLB_ECS_REDIS | A web architecture featuring a caching layer, utilizing SLB, ECS, and Redis. |
| 4 | EIP_SLB_ECS | A public-facing application using an Elastic IP (EIP) bound to an SLB and an ECS backend. |
| 5 | DUAL_ZONE_ECS_SLB | Implements High Availability (HA) across multiple Availability Zones at the compute layer. |
| 6 | DUAL_ZONE_ECS_SLB_RDS | A full-stack HA architecture featuring cross-zone ECS instances and a Primary/Secondary RDS deployment. |

*Table 12.* Cloud Infrastructure Benchmarking Scenarios

**Trace Data Structure.** Each entry in the dataset is a serialized execution trace $\tau = (I, \mathcal{A}, \mathcal{B})$, where:

- **Intent ($I$):** The natural language instruction (e.g., "Create a 2-core ECS in Hangzhou Zone H").

- **Action Sequence ($\mathcal{A}$):** The linear sequence of API calls executed by the agent (e.g., `CreateVpc` $\rightarrow$ `RunInstances`).

- **Blackboard State ($\mathcal{B}$):** The shared context containing resource IDs (e.g., `VpcId`, `SecurityGroupId`) produced by earlier actions and consumed by later ones.

Figure 15 illustrates a sample trace from Scenario S1 (`simple_ecs`). Although the agent executes the actions sequentially (System 2 behavior), the underlying dependencies reveal latent concurrency: `CreateVSwitch` and `CreateSecurityGroup` both depend on `CreateVpc`, but are independent of each other.

F.3.2. TRACE GENERATION PROTOCOL

Execution-derived Cloud-IaC traces are generated as follows: (1) run each cloud-provisioning scenario using the LLM agent environment; (2) record the ordered sequence of successful API calls and blackboard states; (3) project the raw session to

*Table 13.* Glossary of Cloud Infrastructure Terms

| Term | Description |
|---|---|
| **ECS** (Elastic Compute Service) | A web service that provides resizable compute capacity in the cloud (virtual servers), allowing users to launch instances with a variety of operating systems and hardware configurations. |
| **SLB** (Server Load Balancer) | A traffic distribution service that manages high traffic by distributing incoming network requests across multiple ECS instances to ensure high availability and reliability. |
| **RDS** (Relational Database Service) | A managed database service that provides scalable and reliable relational databases (e.g., MySQL, PostgreSQL) without the need for manual hardware provisioning or maintenance. |
| **VPC** (Virtual Private Cloud) | A private, isolated network environment within the cloud where users can configure IP address ranges, subnets, and routing tables to securely manage their resources. |
| **VSwitch** (Virtual Switch) | A virtual networking component within a VPC that connects different cloud resources (like ECS instances) in a specific zone or subnet. |
| **EIP** (Elastic IP) | A static, public IP address designed for dynamic cloud computing, allowing users to mask the failure of an instance or software by rapidly remapping the address to another instance. |
| **Redis** | An in-memory data structure store used as a database, cache, and message broker, often utilized in web architectures to improve performance. |
| **HA** (High Availability) | A system design approach that ensures a certain level of operational performance (uptime) for a higher-than-normal period, often achieved by deploying resources across multiple zones (e.g., Dual Zone). |

*Table 14.* **Cloud-IaC-6 Benchmark Statistics.** $|A|$ and $|E|$ denote nodes/edges in the ground-truth graph. **IP-Cov** measures the diversity of action orderings observed in the dataset.

| ID | Scenario Name | $|A|$ | $|E|$ | n | IP-Cov |
|---|---|---|---|---|---|
| S1 | `simple_ecs` | 5 | 5 | 10 | 100.0% |
| S2 | `slb_ecs_rds` | 12 | 14 | 9 | 12.5% |
| S3 | `slb_ecs_redis` | 9 | 10 | 10 | 53.3% |
| S4 | `eip_slb_ecs` | 9 | 10 | 10 | 43.8% |
| S5 | `dual_zone_ecs_slb` | 7 | 8 | 8 | 40.0% |
| S6 | `dual_zone_..._rds` | 10 | 12 | 7 | 22.2% |

atomic cloud actions using the parsing rule in Appendix D.1; and (4) for controlled IP-Cov experiments, sample additional valid topological orders from the expert dependency graph using randomized Kahn-style frontier sampling.

### F.3.3. EXPERIMENTAL EFFICIENT ENGINE

The experiments in Section 5.5.2 focus on Hybrid mode performance, as this is the most practically relevant regime where partial order inference provides value while maintaining robustness guarantees.

```
# experiment_config.yaml
experiment:
  scenarios: 6        # Cloud-IaC-6
  edge_threshold: 0.5
  eps_values: [0.01, 0.02, 0.03, 0.05]
  ip_cov_targets: [0.6, 0.7, 0.8, 0.9, 1.0]
  total_configs: 120

llm:
  model: qwen3-max
```

```
{
  "trace_id": "T01_qwen-plus_20260104",
  "intent": "Create a 2-core 4G ECS instance in Hangzhou Zone H",
  "action_sequence": [
    { "step": 1, "action": "CreateVpc",
      "output": {"VpcId": "vpc-9517..."} },

    { "step": 2, "action": "CreateVSwitch",
      "params": {"VpcId": "vpc-9517...", "ZoneId": "cn-hangzhou-h"},
      "output": {"VSwitchId": "vsw-191b..."} },

    { "step": 3, "action": "CreateSecurityGroup",
      "params": {"VpcId": "vpc-9517..."},
      "output": {"SecurityGroupId": "sg-0fae..."} },

    { "step": 4, "action": "RunInstances",
      "params": {"VSwitchId": "vsw-191b...", "SecurityGroupId": "sg-0fae..."},
      "output": {"InstanceId": "i-007d..."} }
  ]
}
```

*Figure 15.* **Sample Execution Trace (S1: Simple ECS).** The log captures the linear execution of actions. Note the explicit data dependencies: Step 4 requires outputs from Steps 2 and 3, while Steps 2 and 3 only require Step 1.

```
  temperature: 0.0
  max_tokens: 4096

execution:
  mode: hybrid
  max_workers: 20
  fallback_enabled: true
  timeout_seconds: 300
```

All experiments were conducted on a single workstation with:

- CPU: Apple M2 Max (12 cores)

- Memory: 32 GB

- LLM API: Alibaba Cloud DashScope (qwen3-max)

- Total API cost: approximately $15 USD for all 120 configurations

- Total wall-clock time: approximately 4 hours with 20 parallel workers

### F.3.4. COMPUTATIONAL EFFICIENCY AND SCALABILITY

We report detailed runtime and memory measurements to address practical deployment concerns.

**Runtime Analysis.** Table 15 summarizes the computational cost. Each MCMC run ($10^6$ iterations) completes in approximately **9 minutes** on a single core of an Apple M1 CPU. With 8 parallel workers, the complete experiment suite (120 configurations ) finishes in **5.3 hours wall-clock time**.

Compared to process mining baselines (Inductive/Heuristics Miner), which run in seconds, BPOP trades speed for principled uncertainty quantification and superior peak accuracy. This trade-off is justified for applications requiring high-fidelity structural recovery, such as compliance verification and workflow optimization.

### F.3.5. THE MCMC DETAIL AND RESULTS

Figure 17 illustrates MCMC convergence for a representative run on the `eip_slb_ecs` scenario (Experiment 108) at full trace diversity ($IP\text{-}Cov = 1.0$). The traces demonstrate stable mixing after burn-in, with posterior estimates converging to a noise level of $\rho \approx 0.29$ and a topological depth of $K \approx 3.33$.

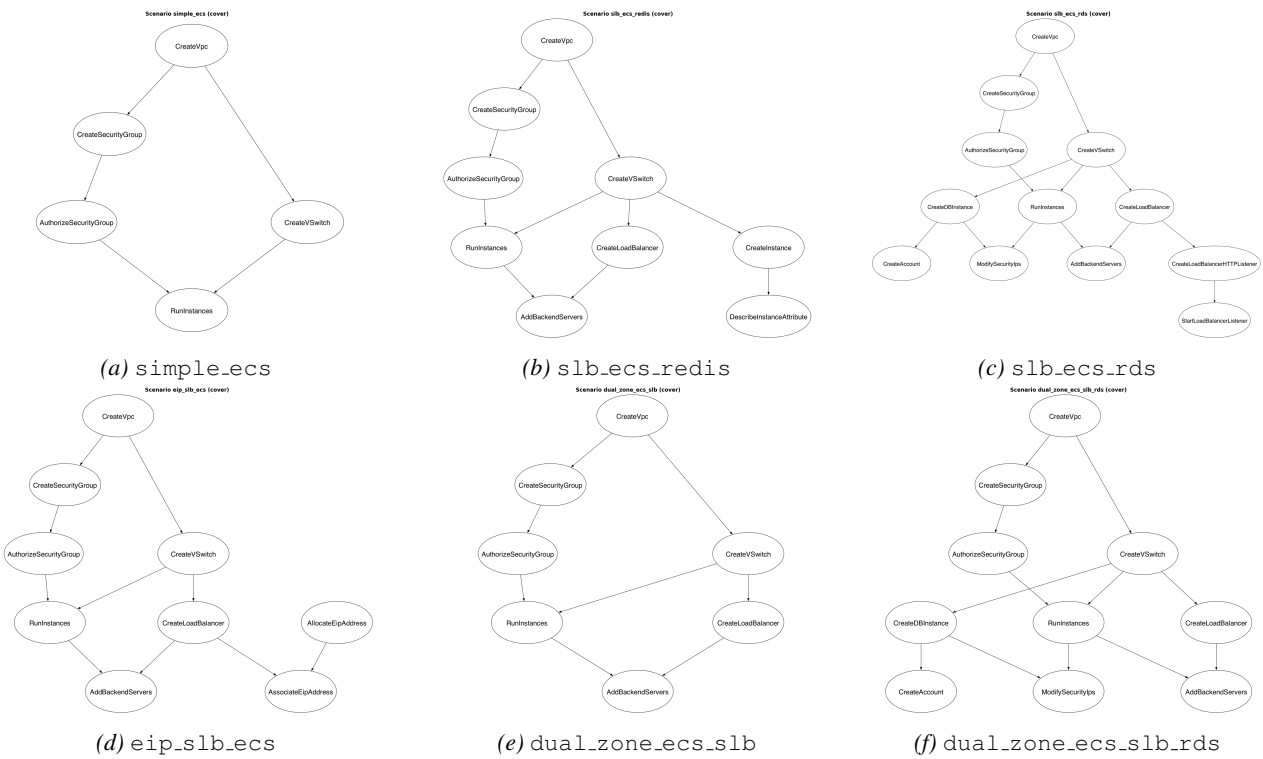

*Figure 16.* Ground-truth action-precedence graphs (covers) for the six Cloud-IaC scenarios. Nodes are cloud API actions; edges denote mandatory precedence constraints.

*Table 15.* Computational cost of BPOP inference.

| Metric | Value |
|---|---|
| MCMC iterations | $10^6$ |
| Parallel workers | 8 |
| Wall-clock time (120 runs) | 5.3 hours |
| Peak memory per run | $<$500 MB |
| Posterior storage (H_trace) | 9.6 MB |

### F.3.6. SENSITIVITY ANALYSIS: POSTERIOR THRESHOLD SELECTION

**Theoretical Intuition vs. Empirical Optima.** The threshold $\alpha = 1/3 \approx 0.33$ is theoretically motivated by a "Three-State" prior. For any pair of nodes $(i, j)$, there are three mutually exclusive relationships: precedence $(i \to j)$, reverse precedence $(j \to i)$, or incomparability $(i \parallel j)$. Under a uniform prior, each state has probability $p = 1/3$. Thus, a posterior probability $\hat{\pi}_{ij} > 1/3$ indicates that the data provides positive evidence for an edge relative to the uniform baseline.

By looking at table 16, we find that a slightly permissive threshold of $\alpha = 0.30$ yields the best performance across all metrics (Edge F1: **0.771**, SHD: **5.7**). The method is robust near the theoretical baseline of $\alpha = 1/3$ (Edge F1: 0.747), validating our three-state intuition. However, performance degrades sharply at $\alpha \geq 0.40$. This indicates that many true dependency edges in sparse workflows carry posterior probabilities in the $[0.30, 0.40)$ range; using a conservative threshold (e.g., 0.50) discards these "weak but real" signals, resulting in false negatives that compromise execution safety.

### F.3.7. OTHER RECOVERY EVALUATION RESULT

**Detailed Feasibility Analysis** Figure 18 provides a scenario-level breakdown of execution feasibility. While the aggregate results in the main text showed a general trend, these plots reveal that baseline failures are often catastrophic in specific complex environments. For instance, in slb_ecs_redis (S3) and eip_slb_ecs (S4), the Heuristics Miner produces graphs that are 100% invalid (0.0 feasibility) at high trace diversity, whereas BPOP maintains near-perfect validity.

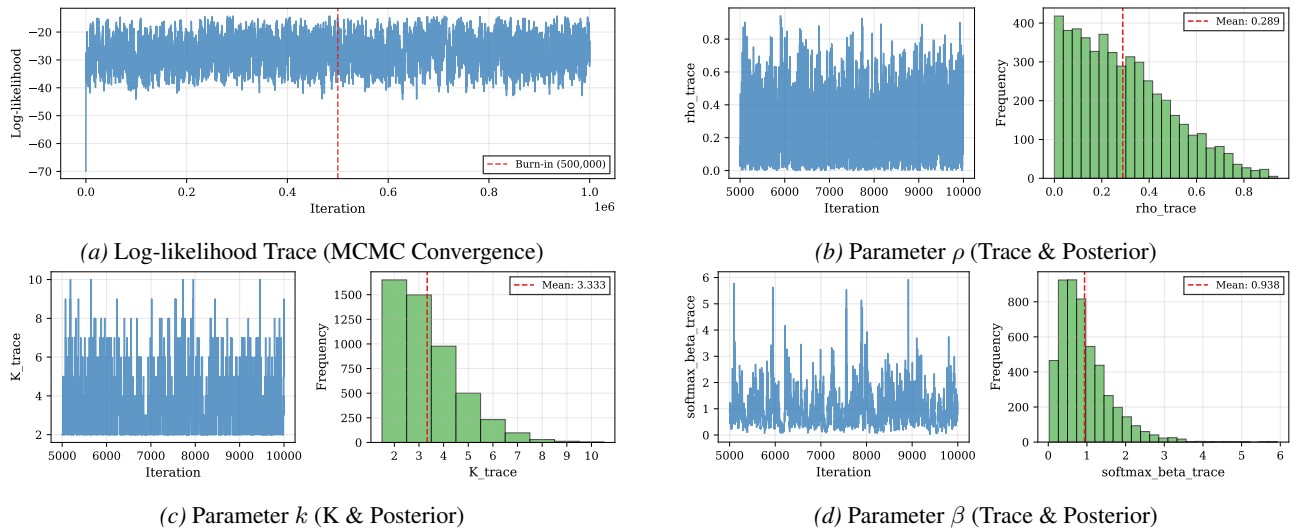

*(a)* Log-likelihood Trace (MCMC Convergence)

*(b)* Parameter $\rho$ (Trace & Posterior)

*(c)* Parameter $k$ (K & Posterior)

*(d)* Parameter $\beta$ (Trace & Posterior)

*Figure 17.* **MCMC Diagnostics (Cloud-IaC-6/`eip_slb_ecs`).** The log-likelihood trace (a) indicates stable convergence after burn-in. Panels (b),(c),(d) show the trace plots and posterior distributions for the latent parameters $\rho$,$k$ and $\beta$ extracted directly from the sampler output.

*Table 16.* Sensitivity Analysis of Posterior Thresholds. While $\alpha = 1/3$ offers strong theoretical justification, $\alpha = 0.30$ is empirically optimal, capturing weak dependencies without introducing noise.

| Threshold | Edge F1 | IP F1 | SHD $\downarrow$ |
|---|---|---|---|
| $\alpha = 0.30$ (Empirical Best) | **0.771** | **0.898** | **5.7** |
| $\alpha = 1/3$ (Theoretical) | 0.747 | 0.893 | 6.2 |
| $\alpha = 0.40$ | 0.665 | 0.856 | 7.0 |
| $\alpha = 0.50$ | 0.514 | 0.801 | 9.3 |
| Marginal Mode | 0.518 | 0.803 | 9.2 |

**Qualitative Recovery at Different Data Regimes** To assess safety in data-scarce regimes, Figure 19 visualizes the recovered graphs at full trace coverage (IP-Cov = 1.0) Figure 20 visualizes the recovered graphs at only partial trace coverage (IP-Cov = 0.6). Even with incomplete data, BPOP recovers the majority of the correct backbone (green edges). While more spurious edges (orange dashed) appear compared to the full-data setting (Main Text Figure), the method successfully avoids the missing edges (false negatives) that would cause runtime failures.

### F.3.8. ADDITIONAL EXECUTION-EFFICIENCY RESULTS

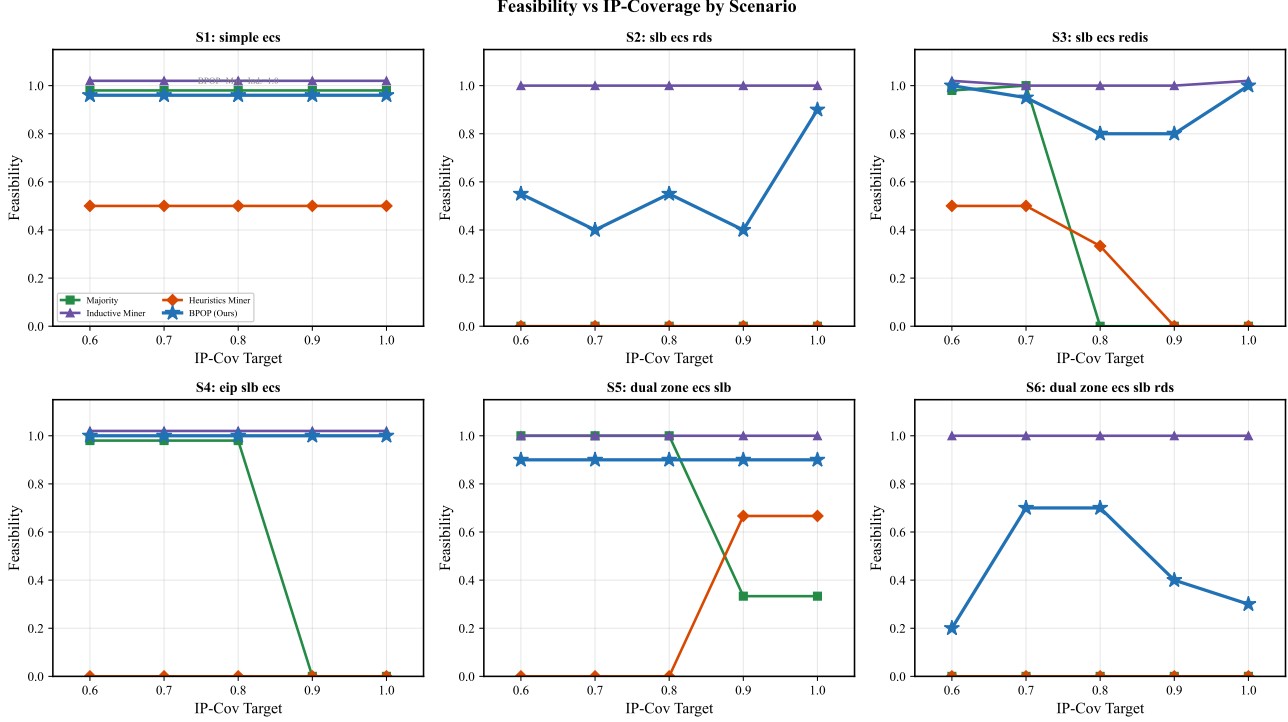

*Figure 18.* **Feasibility vs. IP-Cov by Scenario.** Unlike baselines, which often degrade to 0.0 feasibility in complex scenarios (S3, S4, S6) as trace diversity increases (due to conflicting ordering signals), BPOP maintains high execution validity across all benchmarks.

*Table 17.* BPOP Edge F1 across noise parameter $\epsilon$ and trace diversity (IP-Cov). At high IP-Cov ($\geq 0.9$), performance is stable across all $\epsilon$ values, confirming robustness to the noise parameter.

| IP-Cov | $\epsilon = 0.005$ | $\epsilon = 0.01$ | $\epsilon = 0.02$ | $\epsilon = 0.05$ |
|---|---|---|---|---|
| 0.6 | 0.544 | 0.571 | 0.582 | 0.621 |
| 0.7 | 0.549 | 0.553 | 0.599 | 0.643 |
| 0.8 | 0.642 | 0.645 | 0.702 | 0.722 |
| 0.9 | 0.955 | 0.945 | 0.945 | 0.945 |
| 1.0 | 0.940 | 0.946 | 0.946 | 0.952 |

*Figure 19.* **Qualitative Structure Recovery.** Comparison of BPOP-inferred SOPs against ground truth at high diversity ($IP\text{-}Cov = 1.0$).

*Figure 20.* **Qualitative Structure Recovery at Low Diversity (IP-Cov=0.6).** Even with limited observations (60% coverage of ordering pairs), BPOP correctly identifies the core dependency structure (Solid Green). **Orange Dashed** lines indicate spurious dependencies where the model defaulted to "sequential" due to lack of evidence for concurrency—a safe fallback behavior.

**Example User Case.** This boxed appendix illustrates a representative execution trace comparing *Expert* and *Hybrid* modes on the same cloud provisioning task, highlighting their different responses to execution errors.

---

### Expert vs. Hybrid Reasoning Traces

**User Query.**

**Task:** *Provision a public-facing load-balanced service by allocating an Elastic IP (EIP), binding it to a Server Load Balancer (SLB), and configuring a single Elastic Compute Service (ECS) instance as the backend*

**Expert Mode Reasoning (Failure Case).**
*Reasoning summary.* Expert mode executes the compiled SOP deterministically without re-planning, assuming prerequisite resources (e.g., security groups) already exist. In this trace that assumption is violated, triggering an API error that Expert mode cannot recover from.

**Observed actions (Expert, truncated):**

| Step | Action | Outcome |
|---|---|---|
| 1 | AllocateEipAddress | Success |
| 2 | CreateVpc | Success |
| 3 | AuthorizeSecurityGroup | **Fail** (missing SecurityGroupId) |

**Hybrid Mode Reasoning (Recovery Case).**
*Reasoning summary.* Hybrid mode starts from the same compiled SOP but monitors execution outcomes. On error it triggers LLM-based fallback planning, infers missing prerequisites, and reorders actions before resuming execution.

**Observed actions (Hybrid, truncated):**

| Step | Action | Outcome |
|---|---|---|
| 1 | CreateVSwitch | Success |
| 2 | CreateSecurityGroup | Success |
| 3 | RunInstances (ECS) | Success |
| 4 | CreateLoadBalancer | Success |
| 5 | AddBackendServers | Success |
| 6 | AssociateEipAddress | Success |

**Comparison of reasoning behavior.**

| | Expert Mode | Hybrid Mode |
|---|---|---|
| Reasoning strategy | Deterministic SOP execution | Error-aware replanning |
| Assumptions | Prerequisites already satisfied | Prerequisites inferred dynamically |
| Failure handling | None | LLM-guided recovery |
| Outcome | Execution failure | Successful completion |

**Discussion.** This boxed comparison highlights the complementary roles of the two modes: Expert execution offers low-latency, low-cost runs when the SOP is correct, while Hybrid execution acts as a safety net that guarantees completion under partial structural errors. We report summarized reasoning rather than verbatim chain-of-thought; full internal prompts and hidden reasoning tokens are omitted for clarity and safety.

*Table 18.* **Scenario legend:** S1=simple_ecs, S2=slb_ecs_rds, S3=slb_ecs_redis, S4=eip_slb_ecs, S5=dual_zone_ecs_slb, S6=dual_zone_ecs_slb_rds.

| Scenario | Comp. (%) | Act. /task | Task FB (%) | Act. FB (%) | Calls /task | Tokens /task |
|---|---|---|---|---|---|---|
| S1 | 100.0 | 5.0 | 0.0 | 0.0 | 1.0 | ~**583**[†] |
| S2 | 45.0 | 10.2 | 60.0 | 54.8 | 7.3 | 39,471 |
| S3 | 80.0 | 8.2 | 40.0 | 45.0 | 5.3 | 27,815 |
| S4 | 60.0 | 8.8 | 60.0 | 49.7 | 5.5 | 27,569 |
| S5 | 100.0 | 7.0 | 0.0 | 0.0 | 1.0 | ~**583**[†] |
| S6 | 100.0 | 10.0 | 0.0 | 0.0 | 1.0 | ~**583**[†] |

*Table 19.* LLM Model Sensitivity in Trace Generation (Explore Mode). "Low-level" or high-temperature models (e.g., Qwen-Turbo at $T = 0.5$) struggle to autonomously complete workflows (SR $< 100\%$). However, once the graph is recovered, BPOP's Expert Mode enables these models to execute successfully by offloading reasoning to the engine.

| Model | Temp ($T$) | Success (SR) | Time (s) | Tokens | Rec. |
|---|---|---|---|---|---|
| *High Capability / Low Noise* | | | | | |
| qwen-turbo | 0.3 | **100%** | **74.4** | **40,982** | ⋆⋆⋆ |
| qwen-plus | 0.0 | 100% | 135.7 | 50,652 | ⋆⋆ |
| qwen-plus | 0.3 | 100% | 143.0 | 57,268 | ⋆⋆ |
| qwen3-max | 0.0 | 100% | 122.9 | 59,892 | ⋆⋆ |
| glm-4.7 | 0.3 | 100% | 111.8 | 71,448 | ⋆⋆ |
| *Lower Capability / High Noise (Autonomous Failure)* | | | | | |
| qwen-flash | 0.5 | 83.3% | 101.7 | 54,996 | ⋆ |
| qwen-turbo | 0.5 | 66.7% | 54.8 | 35,315 | ⋆ |
| deepseek-v3.2 | 0.0 | 83.3% | 498.4 | 79,372 | ⋆ |
| kimi-k2 | 0.0 | 83.3% | 626.1 | 98,781 | ⋆ |

