# OpenReview forum: "De-Linearizing Agent Traces: Bayesian Inference of Latent Partial Orders for Efficient Execution"
_ICML.cc/2026/Conference — ICML 2026 regular_

### Official Review · Reviewer_yQKk · 2026-03-12

**Soundness:** 3
**Presentation:** 2
**Significance:** 3
**Originality:** 3
**Overall Recommendation:** 4
**Confidence:** 1

**Summary:**

This paper studies a Bayesian method that handles successful agent traces as noisy linear extensions of the underlying partial order. What they basically suggest is that agent experience should be a reusable structure. What it means is that, they suggest that successful agent traces should not be viewed as linear, and they are just a linearized view of an underlying dependency graph. To my understanding, BPOP is to infer that hidden dependency structure from the given traces then they compile this into an executable graph so that future runs can avoid per step LLN planning. I would like to note that this paper is not directly related to my research area and my guess is merely an educated guess.

**Compliance With Llm Reviewing Policy:**

Affirmed.

**Final Justification:**

I thank the authors for their rebuttal. I will keep my positive score.

**Key Questions For Authors:**

1. I think it would be good to include more background exposure to the paper. Right now, for people not directly involved in this area, it is difficult to follow.

**Limitations:**

yes.

**Strengths And Weaknesses:**

### Strengths

1. I am not able to assess whether experimental settings make sense. But, the results are convincing and it seems that BPOP has significant improvement over the existing baselines such as Majority, inductive or heuristic miners.

2. I think the focus of this paper is also impactful. In my opinion, this is not a widely studied problem but with the more agentic structures we see, the more impact this paper demonstrates.

### Weaknesses

1. Figure 6 shows that with small  IP-Cov Target (such as 0.5). the BPOP has performance less than the baselines provided. Why is that? Also, it seems that sampling rate for IP-Cov Target is too low. We need more granularity to understand what is happening between 0.5 and 0.6.

2. The paper seems to suggest that alpha=1/3 as a theoretically motivated default but the appendix demonstrates that empirical alpha is dependent on the workflow (and fluctuates by dataset). How does alpha chosen in downstream evaluations?

3. BPOP seems to be costly and this presents scalability issues.

---

> ### Author Rebuttal · Authors · 2026-03-27
>
> Thank you for the encouraging review. We are glad the reviewer found the problem impactful and the empirical gains convincing. The core intuition is this: traces such as A-B-C-D and A-C-B-D can be two successful linearizations of the same latent dependency graph; BPOP tries to recover that reusable structure once, and then future runs execute directly on the graph rather than re-planning step by step. We will spell this out more clearly in a revision.
>
> **Q1. The paper would benefit from more background exposure; it is difficult to follow for readers not directly involved in this area.**
> We agree and thank the reviewer for this suggestion. This direction is still relatively new at the intersection of **partial-order theory**, **Bayesian inference**, and **agent execution**, so some concepts may be unfamiliar to readers outside the area. In the revision, we will add a clearer running example earlier in the paper, provide more intuition for the likelihood and execution semantics, and expand the background discussion. For additional intuition: [concrete workflow example](https://anonymous.4open.science/r/icml_rebuttal_po-870F/po_example.png).
>
> **W1. Figure 6 shows that at small IP-Cov targets (e.g., 0.5), BPOP performs worse than some baselines. Why is that? Also, the sampling rate for IP-Cov seems too coarse, especially between 0.5 and 0.6.**
> Low IP-Cov is challenging because few flipped orderings are observed, so concurrency is under-identified from traces. In this regime, BPOP tends to fail conservatively by adding extra edges (over-serializing) rather than omitting critical dependencies. This helps preserve recall of essential prerequisite relations, but can reduce precision and therefore lower the overall F1 score. This is why we also report feasibility in addition to structural recovery. Please see our answer to reviewer skhp W2 and ad3J W2/W7/Q3. We agree that denser sampling between 0.5 and 0.6 would better localize the transition region, although we do not expect it to materially change the overall trend.
>
> **W2. The paper suggests $\alpha = 1/3$ as a theoretically motivated default, but the appendix shows that empirical $\alpha$ depends on the workflow. How is $\alpha$ chosen in downstream evaluations?**
>
> In the downstream Aliyun execution experiments, we use a fixed value of $\alpha$ throughout; it is not tuned per instance or adapted online during execution. The value is included in the paper as a baseline rule of thumb, approximating the Bayes estimator for a loss targeting structure recovery, rather than as a universally optimal choice. It should be close to, but not the same as, the optimal $\alpha$ for feasibility. For that reason, our preferred value of $\alpha$ can vary with workflow characteristics such as graph size and structural complexity: empirically, smaller graphs often favor a larger $\alpha$, whereas larger graphs can favor a smaller $\alpha$. Thus, the practical choice of $\alpha$ need not coincide with the theoretical default. We agree that this distinction is not currently clear enough, and we will revise the text to state it more explicitly.
>
> **W3. BPOP seems costly and may present scalability issues.**
> This point is closely related to the scalability concern raised by **Reviewer skhp (Q2)**; please see our detailed response above. Briefly, the current paper focuses on **single structured tasks** with moderate action spaces, and aims to establish that agent traces can be modeled effectively as latent partial orders with good recoverability and feasibility. BPOP is an offline compilation step, not a per-execution cost. The cost is paid once to infer a reusable graph, and then future runs avoid per-step LLM planning, so the cost is amortized. For larger or longer-horizon settings, a **hierarchical partial-order model** is the natural next step: each subtask can be represented by a smaller partial order, with reusable subtasks drawn from a **shared library**, which will save the cost further.  More scalable inference methods, such as variational inference, are also a promising direction for further reducing runtime.
>
> We also note that the reported MCMC runs were deliberately conservative: we ran them long enough to ensure convergence and collect sufficient posterior samples. Based on the likelihood traces in Figures 13 and 14, the runtime could likely be reduced substantially in practice.

---

> > ### Author Rebuttal · Reviewer_yQKk · 2026-04-03
> >
> > I thank the authors for their rebuttal. I will keep my positive score.

---

### Official Review · Reviewer_ad3J · 2026-03-13

**Soundness:** 2
**Presentation:** 4
**Significance:** 3
**Originality:** 3
**Overall Recommendation:** 5
**Confidence:** 3

**Summary:**

This paper introduces BPOP, a Bayesian framework for learning an executable partial-order workflow from noisy, linearized agent traces. The paper treats each observed trace as a linearized execution of an underlying partial-order workflow. It models the trace as a sequence of choices from the currently feasible actions. To make inference tractable, they introduce a frontier-softmax likelihood instead of summing over all valid linearizations. The method also uses a low-dimensional latent-space representation to define a prior over partial orders. On WFCommons and the new Cloud-IaC-6 benchmark, the paper reports better workflow recovery than process-mining and simple trace-aggregation baselines. It also evaluates a frontier-based executor built from the inferred workflows and reports lower LLM token usage and latency in downstream execution.

**Compliance With Llm Reviewing Policy:**

Affirmed.

**Final Justification:**

Most of my concerns have been addressed. I raised my score accordingly.

**Key Questions For Authors:**

1. How well does the frontier-softmax likelihood reflect real LLM-agent traces, rather than mainly serving as a tractable surrogate?
2. How sensitive is the method to the choice of utility in the frontier-softmax likelihood? Would the results remain similar under simpler alternatives?
3. The IP-Cov analysis suggests that recovery depends on observing enough ordering variation across traces. How robust is the method when repeated traces are highly similar and do not reveal much such variation?
4. How broadly do the authors believe the method generalizes beyond repeated, structured workflow settings?

**Limitations:**

The paper discusses scope limitations such as dependence on trace diversity and restriction to DAG-structured workflows. It would benefit from a discussion of whether the frontier-softmax likelihood is a realistic model of agent traces or mainly a tractable surrogate.

**Strengths And Weaknesses:**

Strengths:

The paper studies a timely problem and formulates it in a meaningful way: it treats repeated agent traces as noisy linearizations of an underlying workflow, with the goal of recovering reusable structure for future execution.

The overall framework is coherent and end-to-end. It connects latent partial-order inference with a frontier-based executor, rather than stopping at structure recovery alone.

A meaningful technical contribution is the frontier-based likelihood, which avoids explicit marginalization over all linear extensions and makes inference computationally manageable.

The empirical evaluation considers both structure recovery and downstream execution behavior, rather than focusing on only one of these aspects.

Weaknesses:

The frontier-softmax likelihood depends on a fairly strong behavioral assumption: among feasible actions, the agent prefers those with high descendant count via the handcrafted utility.

The method appears to depend strongly on having sufficiently diverse traces to reveal the underlying partial order. The paper’s IP-Cov analysis suggests that recovery becomes much harder when repeated traces show little variation in action ordering, which may limit practical applicability.

The evaluated tasks appear relatively specialized and strongly workflow-driven. Because of this, the results seem to support repeated structured workflows more directly than general LLM-agent execution settings.

The related work positioning is not fully clear, especially relative to process mining, workflow discovery, and prior partial-order learning.

Equation (8) includes a parameter τ that is not clearly defined in the surrounding text.

Table 2 reports 1.0 LLM call/task but 0 tokens/task, which is hard to reconcile with Table 7’s metric definition.

The downstream benefits are compelling only once trace diversity is high enough. Table 2 shows a sharp transition: below high IP-Cov, fallback and token usage remain substantial; at high IP-Cov, they collapse. That is a meaningful result, but it also means the method’s practical significance depends heavily on the feasibility of collecting enough diverse successful traces. The current paper does not yet demonstrate that this is easy in real settings, especially since the real Cloud-IaC traces seem relatively low-diversity.

The paper is original overall, but the novelty is moderate. The latent-poset prior builds directly on prior Bayesian partial-order work, and the system contribution is partly an execution wrapper around inferred DAGs. I do think the frontier-softmax idea is nontrivial, but the paper would benefit from isolating more sharply what is genuinely new and what is adaptation.

---

> ### Author Rebuttal · Authors · 2026-03-27
>
> **W1 / Q1 / Q2. Frontier-softmax assumption and sensitivity**.
> We agree that frontier-softmax introduces a behavioral assumption. Our intent is not to claim that all agents literally optimize descendant count, but to use it as a simple and tractable inductive bias: in many structured workflows, prioritizing actions that unlock more downstream work is a reasonable approximation. It should therefore be viewed as a practical surrogate for structure recovery, rather than a literal model of agent policy. In fact we believe the FSM-model is a win on both scalability and model-accuracy. It seems unlikely an agent would select actions weighted by the number of linear extensions. However, a rational agent might well have a bias toward solving the hardest/longest paths first, because it’s a waste of resources to work on the easy part if the hard part fails.
>
> We also explored alternative likelihoods, including a uniform-frontier variant (Q=0) and the earlier Bayesian queue-jump model; comparison results [comparison figure](https://anonymous.4open.science/r/icml_rebuttal_po-870F/po_aliyun_likelihood.png). The main qualitative conclusions remain unchanged, suggesting that performance is driven more by the frontier-based likelihood structure itself than by the exact choice of utility Q. Averaged across all IP-Cov levels, the current BPOP-QSucc variant achieves slightly better structural recovery than BPOP-Uniform (F1: 0.747 vs 0.737). Both frontier-based variants also outperform the earlier Bayesian queue-jump model and are substantially more computationally efficient.
>
> The real agents have heterogeneous preferences, and allowing agent-specific utility functions is an interesting direction, within the framework we set out, but beyond the scope of the current paper.
>
> **W2/W7/Q3. The method appears to depend strongly on high IP-Cov?**
> These points are related to **Weakness 2 raised by Reviewer skhp**; please see our response above. Low trace diversity is an identifiability limit for any trace-only method: if traces exhibit little ordering variation, many incomparable pairs are simply not recoverable from data alone. BPOP tends to fail conservatively by over-serializing rather than omitting prerequisites, which is why we report feasibility in addition to structural recovery. The practical question is then not whether every incomparable pair is recovered, but whether the inferred graph preserves the critical dependencies needed for valid execution.
>
> **W3 / Q4. Scope beyond repeated structured workflows.
> We do not claim that BPOP is a general solution for arbitrary open-ended LLM-agent execution. Rather, it targets settings where agents repeatedly solve similar costly tasks and historical successful traces can be used to infer a reusable dependency structure for future execution, such as cloud provisioning, enterprise SOP automation, and scientific or data-processing workflows.
> This is closely related to Reviewer skhp, Q3(Please refer the answer above): the current framework is DAG-based by design.
>
> **W4. The related-work positioning is not fully clear.**
> This point is closely related to **Reviewer T7Hw (W4)**. Due to space constraints, the related-work discussion is compressed; We will sharpen the positioning relative to process mining, workflow discovery, and prior Bayesian partial-order learning.
>
> **W5. Eq8 includes  $\tau$ not defined**
> This is a typo from an earlier version of the model: it should simply be deleted from the the LHS of  Eq. (8).
>
> **W6. Table2 reports 1.0 LLM call/task but 0 tokens/task, hard to reconcile with Table7’s metric def.**
> The “0 tokens” entry reflects a bookkeeping oversight; the intent-parsing tokens (approximately **580 tokens/task**) were tracked separately from reasoning tokens and were inadvertently omitted from the aggregate reported in Table 2. We will correct the table in the revision to report approximately **580 tokens/task** for Expert mode, all from intent parsing and **0 reasoning tokens**. The main conclusion is unchanged, since Explore mode is dominated by step-by-step CoT reasoning (approximately 62,700 tokens/task).
>
> **W8. The novelty is moderate.**
> We do not claim that all components are entirely new: the latent-poset prior builds on prior Bayesian partial-order work. That said, we believe the novelty in introducing a observability framework for agent workflows. By recovering an interpretable latent dependency structure from successful traces, BPOP makes agent execution more transparent and therefore supports not only compiled execution, but also workflow understanding, root-cause analysis, efficiency analysis, and optimization.
> New contributions are: (i) the frontier-softmax likelihood for tractable inference from noisy linearized agent traces, (ii) the formulation of agent traces as latent partial-order workflows, (iii) the recoverability/feasibility analysis, and (iv) compilation of inferred structure into an executable graph for downstream efficiency gains.

---

> > ### Author Rebuttal · Reviewer_ad3J · 2026-04-04
> >
> > I thank the authors for their response. I will raise my score accordingly.

---

### Official Review · Reviewer_T7Hw · 2026-03-13

**Soundness:** 3
**Presentation:** 2
**Significance:** 3
**Originality:** 4
**Overall Recommendation:** 5
**Confidence:** 1

**Summary:**

This paper tackles the inefficiency of sequential execution in LLM agents by introducing **Bayesian Partial Order Planning (BPOP)**, a framework that uncovers latent dependency structures from observed execution traces to enable parallel task execution. The approach represents agent workflows as partial orders rather than linear sequences, leveraging Bayesian inference to identify true causal dependencies from historical data. To handle the computational challenge of linear extension counting, the authors propose a tractable frontier-softmax likelihood, reducing complexity to (O(|≻h| + T · |A|) ) instead of exponential. The method is validated on scientific workflows (WFCommons) and cloud infrastructure tasks (Aliyun Cloud-IaC-6), demonstrating substantial efficiency improvements, cutting LLM calls from step-by-step execution to near-constant, while ensuring correctness through a tri-modal execution system with Expert, Hybrid, and Explore modes.

**Compliance With Llm Reviewing Policy:**

Affirmed.

**Final Justification:**

My previous concerns have been largely addressed. The paper is supported by rigorous mathematical analysis, and its strengths outweigh its weaknesses. Accordingly, I have increased my score.

**Key Questions For Authors:**

1. Why focus primarily on process mining baselines rather than modern scheduling approaches? How would BPOP compare to recent learning-based scheduling methods or other agent efficiency approaches like caching and memoization?

**Limitations:**

Yes.

**Strengths And Weaknesses:**

**Strengths:**
### Soundness
- The mathematical formulation is rigorous, with the frontier-softmax likelihood representing a genuine theoretical contribution that makes Bayesian inference over partial orders computationally tractable
- Strong empirical results: 0.95 Edge-F1 in structural recovery with substantial computational efficiency gains
### Presentation
- Clear progression from motivation through technical details to empirical validation
- Effective visualizations (Figure 1's architecture comparison, Figure 5's recovery results)
- Well-structured exposition with formal problem formulation
### Significance
- Addresses the genuine "perpetual intern syndrome" bottleneck in current LLM agents
- Broad applicability across domains (scientific computing, cloud infrastructure) with clear enterprise value
- Significant practical impact: 5.3-hour wall-clock completion time with substantial token cost reduction
- Framework could influence future research in agent efficiency and parallel execution
### Originality
- Novel application of Bayesian partial order inference to agent execution optimization
- Creative solution converting intractable combinatorial problem into tractable sequential choice process

**Weaknesses:**
### Presentation
- Dense mathematical notation at the intersection of partial order theory, Bayesian inference, and execution semantics may challenge readers
### Significance
- Limited scope due to DAG restriction, excluding workflows with legitimate cycles or iterative patterns
- Scalability concerns for very large action spaces (O(|A|²) memory requirements)
### Originality
- Limited discussion and engagement with related workflow scheduling and causal discovery literature

---

> ### Author Rebuttal · Authors · 2026-03-26
>
> Thank you for the positive assessment and the helpful question about baseline choice and scope.
> **W1 (Presentation). Dense mathematical notation**
> We agree that the paper is mathematically dense, and we thank the reviewer for highlighting this. Because this is our first paper in this direction, we intentionally included more detail on the modeling assumptions and theoretical foundations, since a central goal of the paper is to establish partial orders as a principled framework for representing agent traces. In the revision, we will streamline notation in the main text, provide more intuition for the likelihood and execution semantics, and move some technical derivations to the appendix where possible.
>
> **W2 (Significance 1- DAG). Limited scope due to DAG restriction, excluding workflows with legitimate cycles or iterative patterns.**
> This point is closely related to **Q3** raised by **Reviewer skhp**. The current framework is intentionally designed for convergent procedural tasks, where a DAG is not only a restriction but also a practically useful representation because it improves observability, debugging, root cause analysis and workflow optimization.
>
> **W3 (Significance 2- Scalability). Scalability concerns for very large action spaces (\(O(|A|^2)\) memory requirements).**
> We believe there may be a slight misunderstanding here: In our current setting, memory is not the main real bottleneck. Although the implementation uses an \(O(|A|^2)\) graph representation, the main stored object is a **binary dependency matrix**, which can be represented compactly (e.g., via bit-packing), so the practical memory footprint remains modest at the workflow scales considered in this paper. More broadly, our target regime is **single structured tasks** with moderate action spaces, rather than extremely large open-ended action sets.
>
> **W4 (Originality). Limited discussion and engagement with related workflow scheduling and causal discovery literature.**
> We agree that the related-work positioning can be sharpened. In the revision, we will more clearly distinguish BPOP from three adjacent lines of work: (i) process mining / workflow discovery from event logs, which also infer process structure from traces; (ii) workflow scheduling, which typically assumes the workflow DAG is already known and optimizes execution on top of it; and (iii) causal / graphical-model structure learning, including partial-order-based Bayesian-network discovery, which studies probabilistic inference over latent DAG structure from observational data. Our setting differs in that BPOP infers an executable dependency graph from noisy agent traces and uses it not only for execution, but also for agent observability: workflow understanding, root-cause analysis, efficiency analysis, and workflow optimization.
>
> We will expand section 6 accordingly and cite representative work from process mining and partial-order process discovery (e.g., van der Aalst et al., 2003; Leemans et al., 2023), workflow scheduling on known DAGs (e.g., da Silva and Gabriel, 2020), graphical-model and causal structure learning (e.g., Niinimäki et al., 2016; Squires and Uhler, 2023), mutation-order inference (e.g., Gerstung et al., 2011), and recent partial-order methods for LLM response ranking (e.g., Wang et al., 2024).
>
> **Q1. Why focus primarily on process-mining baselines rather than modern scheduling approaches? How would BPOP compare to recent learning-based scheduling methods or other agent-efficiency approaches such as caching and memoization?**
> Our primary baselines are process-mining / workflow-discovery methods because the core problem in this paper is not scheduling a known workflow, but inferring a latent executable dependency graph from successful traces. Process-mining methods are therefore the closest apples-to-apples baselines: like BPOP, they take execution traces as input and attempt to recover reusable workflow structure.
>
> By contrast, most scheduling methods—heuristic or learning-based—assume the DAG (and often resource constraints or execution costs) is already known, and then optimize execution on top of that graph. In this sense, scheduling is downstream and complementary to our problem rather than a direct substitute for it. Once BPOP infers a workflow, more advanced schedulers could be applied on top of the inferred graph as a natural extension.
>
> Caching and memoization are similarly orthogonal to our contribution. They reduce repeated computation for individual substeps, whereas BPOP reduces repeated deliberation by compiling a reusable dependency structure from traces. The two approaches can be combined. We will revise the paper to make this distinction clearer and to better position BPOP relative to workflow discovery, scheduling, causal discovery, and other agent-efficiency methods.

---

> > ### Author Rebuttal · Reviewer_T7Hw · 2026-04-02
> >
> > I appreciate the authors' response, and my concerns are mostly addressed so I will increase the score accordingly.

---

### Official Review · Reviewer_skhp · 2026-03-18

**Soundness:** 3
**Presentation:** 2
**Significance:** 3
**Originality:** 3
**Overall Recommendation:** 4
**Confidence:** 3

**Summary:**

This paper introduces BPOP (Bayesian Partial Order Panning), a probabilistic framework for inferring latent dependency structure from noisy linearized traces (sequence of actions for e.g. LLM agent execution traces). Key idea on which this paper builds is that agent logs are noisy linearizations of an underlying partial order over actions, and if we can recover this partial order, it can be leveraged to parallelize executions. The key technical contribution of the paper is to device an efficient MCMC based inference via a tractable frontier-softmax likelihood which avoids the #P-hard marginalization over linear extensions. The inferred partial order is used to replace the per-step LLM reasoning with GEE (graph execution engine) based deterministic SOP. Experiments across WFCommons scientific workflow and  heterogeneous LLM-generated traces (on cloud provisioning tasks) show that BPOP recovers dependency structure more accurately than trace-only and process-mining baselines, leading to substantial reductions in token usage and execution time.

**Compliance With Llm Reviewing Policy:**

Affirmed.

**Key Questions For Authors:**

1. Graph 1 is not clear - can you make it more descriptive or give actual example of latent structure and traces.

2. Is the method scalable? How does MCMC convergence scale with number of nodes in the underlying latent structure?

3. Can the framework be extended beyond DAG structure?

**Limitations:**

Authors have included a limitation section.

**Strengths And Weaknesses:**

**Strengths**

1. Paper addresses an important problem of identifying underlying latent dependency structures from linearized traces. It has applications beyond LLM traces as well.

2. The solution methodology (BPOP) and technical contribution and the efficient MCMC based inference via a tractable frontier-softmax likelihood is also novel. In addition, it also gives the ability to formulate executable SOPs through GEE.

3. Experimental results show strong improvement over the baselines (Fiigures 4 -8).  In addition, the empirical observation that there is a sharp phase transition in BPOP's performance depending on IP-cov (Table 2) is quite interesting.


**Weaknesses**

1. Paper is a bit hard to follow. Following suggestions might help in improving the paper writing:

     a. Include an example (of latent dependcy structure and few linearized traces) in the introduction or problem formulation.

     b. Make this example as a running example for the rest of the paper to provide exammples and anecdotes.

     c. Explain the methodology broadly through a graph or workflow, and then go into the details of it.

2. Another concern is that BPOP seems to perform well only with high IP-cov, how reasonable is to assume high IP-cov in realistic settings?

---

> ### Author Rebuttal · Authors · 2026-03-26
>
> Thank you for the positive assessment and for the helpful suggestions on presentation.
>
> **W1 / Q1. The paper is somewhat hard to follow, and Figure 1 could be more descriptive.**
> We agree that the presentation can be made easier to follow. In the revision, we will add a compact running example early in the introduction/problem formulation and reuse it throughout the method section. For example, a latent dependency such as Create VPC -> {Create subnet, Create security group} -> Launch instance yields multiple valid traces depending on the order of the middle two actions; BPOP aims to recover that partial order from such linearizations.  Figure 1 is intended primarily as a **motivation/system figure**, showing the high-level idea that we learn a latent dependency graph from observed traces and then use it for graph-based execution, rather than as a detailed methodology figure.  For additional intuition, we also provide an external illustration here: [Concrete workflow example](https://anonymous.4open.science/r/icml_rebuttal_po-870F/po_example.png), showing multiple observed traces, the recovered latent dependency graph, and the resulting compiled execution pipeline.
>
> **Weakness 2. Dependence on high IP-Cov.**
> IP-Cov in Table 2 is a controlled identifiability diagnostic, not a deployment assumption. Its role is to show when the full latent partial order is recoverable in principle. In realistic low-diversity settings, many incomparable pairs are simply not identifiable from traces alone, but exact structural recovery is not the most practically important criterion. For this reason, we also report feasibility as a complementary metric. Feasibility measures whether the inferred graph preserves the critical dependency relations needed for valid execution, even when the data are insufficient to identify all incomparable pairs. Thus, our goal is not only high recoverability when IP-Cov is high, but also effective and safe execution when IP-Cov is low. As shown in Table 1, the method still achieves high feasibility with limited trace diversity, meaning it can often provide a usable execution graph even when full structural recovery is not possible. This is the real win.
>
> Low IP-Cov is fundamentally an identifiability limit for any trace-only method(we never know the true graph), not something specific to BPOP.
>
> **Q2. Is the method scalable? How does MCMC convergence scale with the number of nodes?**
> In the current paper, we demonstrate empirical feasibility up to approximately **n = 50** actions in the second experiment, which already covers many practical structured workflows in our target setting. The key reason is the frontier-softmax likelihood: it avoids the #P-hard marginalization over linear extensions and makes each likelihood evaluation tractable. It's a  improvement compared to previous bayesian partial order model for ranking. More broadly, this work focuses on **single structured tasks**, where the action space is typically moderate. For substantially longer traces, however, a **hierarchical partial-order model** is likely the more natural extension, since such traces often consist of multiple subtasks rather than a single flat workflow. As a first paper on Bayesian agent-trace modeling with partial orders, the present work is primarily aimed at validating the core formulation and showing strong recoverability and feasibility in the intended workflow regime.
>
> **Q3. Can the framework be extended beyond DAG?**
> We do not want to overclaim that the current framework extends beyond DAG structures, and this limitation is already acknowledged. The DAG assumption is not only a restriction but also a practically useful design choice for the setting we study. DAGs provide a clear and interpretable dependency structure, which improves observability, facilitates debugging, and supports workflow optimization and execution-strategy design. These properties are especially valuable in practical agent systems, where the learned workflow should be inspectable and operationally reliable rather than treated as a black box.
>
> As clarified, BPOP is intended for convergent procedural tasks, where agents interact with a finite set of tools/APIs and execution is governed by relatively stable dependency semantics, rather than fully open-ended planning. Within this scope, the method applies to settings such as Cloud-IaC, enterprise SOP automation, CI/CD pipelines, and data-processing workflows. We also validate the framework in two substantially different domains—cloud provisioning and scientific workflows—which supports applicability beyond a single setting. In our offline setting, traces can often be preprocessed to remove loops caused by failures, retries, or reconsideration, yielding clean traces suitable for the model. Extending the framework to handle genuinely cyclic or iterative behaviors more directly is an important direction for future work.

---

> > ### Author Rebuttal · Reviewer_skhp · 2026-04-04
> >
> > Thanks authors for their detailed response. Most of my concerns are addressed.
> >
> > I will keep my positive score (4).

---

### Decision · Program_Chairs · 2026-04-30

**Decision:**

Accept (regular)

**Comment:**

The reviewers had an overall positive impression of this work.  In particular, the reviewers noted that:
1) The frontier-softmax likelihood is a solid theoretical contribution that makes Bayesian inference over partial orders computationally tractable.
2) The empirical evaluation, which considers both structure recovery and downstream execution behavior, shows significant improvement over the existing baselines.

The primary weaknesses identified by the reviewers were
1) The frontier-softmax likelihood makes the behavioral assumption that "among feasible actions, the agent prefers those with high descendant count via the handcrafted utility."
2) Scalability issues (in particular in regards to memory requirements).
3) Novelty is limited in a sense that this builds directly on a line of existing work.  Could add additional citations to conncent this work to the broader community.

Overall, I feel that theoretical and empirical contributions make this a worthwhile contribution.